# Learning Task-Agnostic Representations through Multi-Teacher Distillation

**Philippe Formont***
Universite Paris-Saclay- ETS Montreal
Mila - Quebec AI Institute
LIVIA- ILLS

**Maxime Darrin***
McGill University- Universite Paris-Saclay
Mila - Quebec AI Institute
ILLS

**Banafsheh Karimian***
ETS Montreal
ILLS - LIVIA

**Jackie CK Cheung**
McGill University
Mila - Quebec AI Institute

**Eric Granger**
ETS Montreal
ILLS - LIVIA

**Ismail Ben Ayed**
ETS Montreal
ILLS - LIVIA

**Mohammadhadi Shateri**
ETS Montreal
LIVIA

**Pablo Piantanida**
CNRS - CentraleSupelec - Universite Paris-Saclay
ILLS - Mila - Quebec AI Institute

## Abstract

Casting complex inputs into tractable representations is a critical step across various fields. Diverse embedding models emerge from differences in architectures, loss functions, input modalities and datasets, each capturing unique aspects of the input. Multi-teacher distillation leverages this diversity to enrich representations but often remains tailored to specific tasks. In this paper, we introduce a task-agnostic framework based on a "majority vote" objective function. We demonstrate that this function is bounded by the mutual information between student and teachers' embeddings, leading to a task-agnostic distillation loss that eliminates dependence on task-specific labels or prior knowledge. Our evaluations across text, vision models, and molecular modeling show that our method effectively leverages teacher diversity, resulting in representations enabling better performance for a wide range of downstream tasks such as classification, clustering, or regression. Additionally, we train and release state-of-the-art embedding models, enhancing downstream performance in various modalities.

## 1 Introduction

Transforming complex inputs into tractable representations is crucial for numerous applications across different domains, from natural language processing (Li & Li, 2023; Pimentel et al., 2023), computer vision (Kubota et al., 2024; Bhalla et al., 2024) to bioinformatics (Morgan, 1965; Wang et al., 2022a). This is done using embedders, often large pretrained models (Touvron et al., 2023; Jiang et al., 2023), that project objects (image, text, molecules, . . . ) into numerical representations, enabling various downstream tasks (Murphy, 2013; Vilnis & McCallum, 2015).

Variations in model architecture, training paradigms (e.g., unsupervised vs. supervised), and objective functions (e.g., masked language modeling and contrastive learning) result in embedders that capture different aspects of the same input. To leverage this diversity, a common practice is to combine them into a single model through multi-teacher Knowledge Distillation (KD) (Zhang et al., 2023).

---

*Equal Contribution

39th Conference on Neural Information Processing Systems (NeurIPS 2025).

Not only are these methods cost-effective at inference time (Hinton et al., 2015; Frosst & Hinton, 2017), they are also extremely useful to compress knowledge from larger models into smaller ones for resource-constrained environments (Pan et al., 2022; Wang et al., 2023; Zhang et al., 2023), or mend the weights of models whose architectures have been altered (Muralidharan et al., 2024). Most existing approaches, however, focus on single-task distillation. In this setting, the student model either learns to mimic teacher representations for a specific task (Dvornik et al., 2019), or the distillation process is explicitly paired with task-specific information. While effective, such methods cannot be used for or generalized to unseen tasks, requiring a new distillation process to be performed for every new task. **Our goal is to learn a highly informative representation that retains maximal utility across a wide range of downstream tasks.** In other words, we aim to maximize information density within a single representation, enabling general-purpose adaptability without sacrificing performance.

Task-agnostic multi-teacher distillation aims to compress teacher representations into a single student embedder, such that the student representation captures as much information as all the teachers combined. To our knowledge, few works address task-agnostic distillation from multiple teachers. Existing approaches often rely on mean squared error (MSE) loss and cross-encoder heads (Navaneet et al., 2022), which can be unstable in high-dimensional spaces (Farebrother et al., 2024).

To overcome these limitations, we introduce a novel task-enabling setting to task-agnostic multi-teacher distillation. Our goal is to develop representations that capture the maximum amount of information about the data distribution, ensuring their applicability to a wide range of tasks, even in the absence of prior knowledge about those tasks. We train the student model to learn representations that, when applied to downstream tasks, generate predictions consistent with the majority of predictions from the teachers' representations. This approach allows our method to leverage the collective knowledge of the teachers' ensemble. To achieve this, we introduce an ensembling loss that measures the agreement between the Bayesian predictor based on the student's embeddings and the Bayesian predictors based on the teachers' embeddings. We show that this loss can be bounded independently of the task, using the conditional differential entropy of the teachers' embeddings given the student's output, thus providing a task-agnostic student-teacher reconstruction loss.

**Contributions.** In this study, we investigate the following research question: **How can the knowledge from multiple large embedding models be effectively distilled and integrated into a smaller one to produce a more general-purpose representation?** Our main contributions are threefold:

1. **A task-enabling setting.** We frame the multi-teacher distillation problem in a task-enabling setting, in which we study the relationship between the Bayes classifiers obtained from the students and the teachers' embeddings. We prove a simple, yet powerful result: the conditional entropy of the teachers given the student's output controls the probability of the student's Bayesian predictor disagreeing with the teachers' for any task.

2. **A tractable implementation.** We leverage a recent differentiable high-dimensional Gaussian-Mixture based estimator of the differential conditional entropy to formulate an information-theoretic loss. This loss maximizes the mutual information between the student and all teachers, resulting in a principled, task-agnostic distillation objective.

3. **High-quality generalized embedders.** Our method enhances distillation capabilities across three application domains: molecular modeling, natural language processing and computer vision. We release trained students achieving competitive performance on a wide range of downstream tasks, e.g., classification, regression, clustering, and sentence similarity.

## 2 Related Work

**Task-oriented distillation.** KD is widely used for transferring knowledge from one or a set of teachers to a student model (Gou et al., 2021) to improve the performance of the student on a given task (Zhang et al., 2019; Yim et al., 2017). This is typically done by transferring logits (Sun et al., 2024); *i.e.* the models' output, features (Wang et al., 2023; Sarkar & Etemad, 2024), relational information (Dong et al., 2024, 2021), or a mixture of them (Liu et al., 2021a). Similarly, (Qiu et al., 2024) uses a regularization term to distill the task-relevant information from the large teacher to the small student. We depart from these methods by focusing on distilling task-agnostic representations.

**Task-oriented multi-teacher distillation.** A common method for multi-teacher KD is averaging the teachers' logits and transferring the result to the student (Dvornik et al., 2019; Hinton et al., 2015).

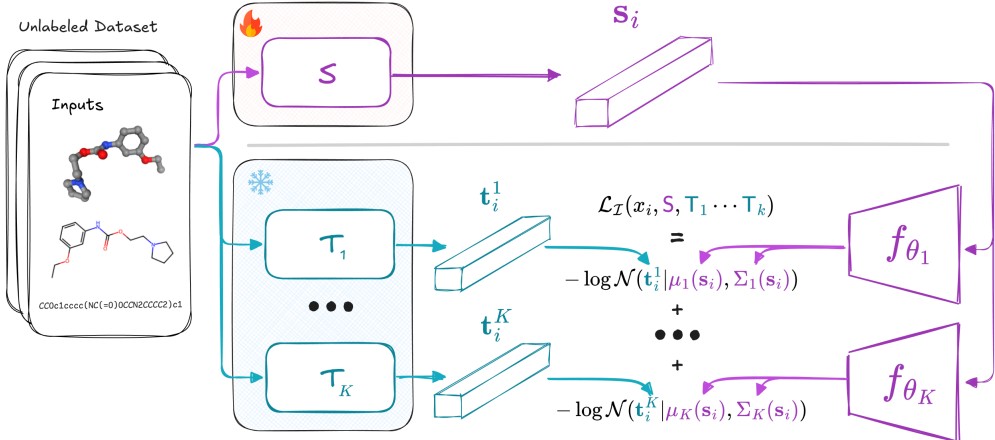

Figure 1: **Unsupervised training of our student through task-agnostic distillation.** The student embedder $S$ is trained to minimize the negative log-likelihood of multiple teachers' outputs conditioned on the student's predictions. During this multi-teacher distillation procedure, both the student's weights and those of the teacher-specific Gaussian kernels $\{f_{\theta_k}\}_{k \leqslant K}$ are updated in an end-to-end fashion. Post-training, we discard the Gaussian kernels and evaluate the student embedders by freezing their weights and training a feed-forward network on their embeddings for an unseen dataset.

However, this approach is not ideal when the performance of the teachers is uncertain. Alternative methods include using gate networks (Zhu et al., 2020), reinforcement learning agents (Yuan et al., 2020), and other methods (Ma et al., 2024a; Borza et al., 2022; Zhang et al., 2023) to perform teacher selection or evaluation. Due to challenges in distilling knowledge among diverse architectures, multi-teacher KD research mainly focuses on logit distillation. Other techniques were also explored, such as multi-teacher feature ensemble (Ye et al., 2024), contrasting feature distillation (Li et al., 2024), and cosine similarity-based methods for various tasks (Ma et al., 2024b; Aslam et al., 2024, 2023). Ensemble-based methods have also been proposed to mitigate over-smoothing and leverage teacher diversity, such as by aggregating structured predictions before distillation (Shayegh et al., 2024). Although successful, most multi-teacher feature distillation methods remain oriented to only one or a few tasks.

**Task-agnostic and self-supervised features distillation.** To the best of our knowledge, few works address task-agnostic representation distillation. Several approaches assume strong limitations, such as requiring the student to have the same architecture as the teachers (Liang et al., 2023; Xu et al., 2022b), or requiring fine-tuning the teachers to then distill their representations (Liu et al., 2023). Other methods induce requirements on the students, limiting their extension to a general multi-teacher setting. Notably (Gao et al., 2022) relies on vision-specific data augmentation, RoB (Duval et al., 2023) focuses on the distillation of joint-embedding approaches, AttnDist (Wang et al., 2022b) is only applicable to single teacher, (Song et al., 2023) need the teacher and student to have the same architecture, and SEED (Fang et al., 2021) requires the student and the teacher to have the same embedding dimension. Finally, CompRess (Abbasi Koohpayegani et al., 2020) introduced a distillation method ensuring that the embeddings of the student and the teacher encode a similar nearest-neighbor graph, which would be unstable in a multi-teacher setting. Other approaches such as contrastive learning (Feng et al., 2024; Liu et al., 2022; Xu et al., 2022a) focus on distilling relational relationships between the samples, such as nearest neighbors preservation (Noroozi et al., 2018) or angle preserving distillation(Park et al., 2019a). SimReg (Navaneet et al., 2022), however, trains the student jointly with cross-encoding heads to directly reconstruct the teacher's features using an MSE loss.

**Interval estimation.** While SimReg performs its distillation through pointwise estimation with MSE, it is well known in the reinforcement learning literature that these standard regression methods are difficult to train (Farebrother et al., 2024). On the other hand, replacing traditional regression scheme by maximum-likelihood training of Gaussian kernels appears to be more stable (Stewart et al., 2023) and effective in Value learning (Bellemare et al., 2017). We extend this idea in the

context of embedder distillation by using Gaussian kernels to estimate the conditional distribution of the teachers' embeddings given the student embedding and show that it is directly connected to maximizing the mutual information between the student and the teacher.

## 3 Distilling Representation Through Gaussian Kernels

We denote the input space by $\mathcal{X}$ and the corresponding input distribution by $P_\mathbf{X}$. We assume we have access to a dataset $\mathcal{D} = \{\mathbf{x}_i\}_{i=1}^n$, where samples are drawn i.i.d. according to $P_\mathbf{X}$. We consider a set of $K$ different teacher embedders, $\mathsf{T}_k : \mathcal{X} \to \mathbb{R}^{d_k}$, for $k \in \{1, \ldots, K\}$, each mapping inputs to potentially different embedding spaces of dimension $d_k$.

### 3.1 From a task-oriented setting to a task-agnostic loss

Our goal is to train a representation model capable of effectively handling any downstream task, by leveraging diverse representations from diverse pretrained teachers (Figure 1). To do so, we first measure the agreement between the student's Bayes classifier and the teachers' for any given task. First, we demonstrate that it can be bounded by the conditional entropy of the teacher's embedding given the student's, which does not depend on the considered task.

Let us consider a task characterized by a target set $\mathcal{Y}$ of discrete concepts and the feature space $\mathcal{X}$ with joint probability measure $P_\mathbf{YX} \in \mathcal{P}(\mathcal{Y} \times \mathcal{X})$. For every projection of the features through the different teachers, the Bayes decision rule is given by $c_{\mathsf{T}_k}^* \triangleq \arg\max_{c:\mathbb{R}^{d_k} \to \mathcal{Y}} \mathbb{E}_{\mathbf{X},\mathbf{Y}} \left[ \mathbb{1} \left[ c(\mathsf{T}_k(\mathbf{X})) = \mathbf{Y} \right] \right]$ and for the student: $c_\mathsf{S}^* \triangleq \arg\max_{c:\mathbb{R}^d \to \mathcal{Y}} \mathbb{E}_{\mathbf{X},\mathbf{Y}} \left[ \mathbb{1} \left[ c(\mathsf{S}(\mathbf{X})) = \mathbf{Y} \right] \right]$.

Our goal is to minimize the probability that the student's Bayesian classifier deviates from the predictions of the teachers'. This approach has been shown to enhance performance in most cases by reducing both bias and variance, while improving robustness and generalizability (Dietterich, 2000; Scimeca et al., 2023; Allen-Zhu & Li, 2020; Theisen et al., 2024). In other words, we aim to minimize the probability that the student's decision differs from that of each teacher:

$$\mathcal{L}^*(\mathbf{X}, \mathbf{Y}, \mathsf{S}, \mathsf{T}_1, \ldots, \mathsf{T}_K) = \frac{1}{K} \sum_{k=1}^K \underbrace{\Pr\left( c_\mathsf{S}^*(\mathsf{S}(\mathbf{X})) \neq c_{\mathsf{T}_k}^*(\mathsf{T}_k(\mathbf{X})) \right)}_{\substack{\text{Probability that the student Bayesian classifier's} \\ \text{output is different from the } k^{\text{th}} \text{ teacher's}}} . \tag{1}$$

where the loss depends on the joint distribution $(\mathbf{X}, \mathbf{Y})$, through the definition of the Bayesian classifiers.

We leverage recent results on the performance of the Bayes classifiers to bound the probability of getting two different outcomes using the Bayes classifiers operating on two different projections of the input space.

**Proposition 3.1** (Darrin et al. (2024)). *Let $C_{\mathsf{T}_k} = c_{\mathsf{T}_k}^*(\mathsf{T}_k(X))$ and $C_\mathsf{S} = c_\mathsf{S}^*(\mathsf{S}(t))$ denote the outcome of the Bayes classifier observing the output of the teacher $\mathsf{T}_k$ and the student $\mathsf{S}$ on a given task $Y$, respectively.*

$$\Pr\left( C_\mathsf{S} \neq C_{\mathsf{T}_k} \right) \leqslant 1 - \exp\left( -h\left( \mathsf{T}_k(X) | \mathsf{S}(X) \right) \right)$$

**Corollary 3.2** (Training objective). *By applying Prop. 3.1 to Eq. 1 for any given joint distribution $P_\mathbf{XY}$, we have*

$$\mathcal{L}^*(\mathbf{X}, \mathbf{Y}, \mathsf{S}, \mathsf{T}_1, \ldots, \mathsf{T}_K) \leqslant 1 - \exp\left( - \underbrace{\frac{1}{K} \sum_{k=1}^K h(\mathsf{T}_k(\mathbf{X}) | \mathsf{S}(\mathbf{X}))}_{\text{Negative log likelihood}} \right). \tag{2}$$

*This corollary directly follows from the concavity of $t \to 1 - \exp(-t)$ (see Appendix A).*

*Remark* 3.3. This bound over our ideal loss $\mathcal{L}^*$ is independent of the specific task and depends solely on the conditional entropy of the teacher embeddings given the student embeddings. Therefore, optimizing the student to minimize this loss provides a task-agnostic approach to aligning its Bayesian classifier predictions with the ensemble of teachers' predictions, regardless of the downstream task.

## 3.2 Student training

**Estimation of the conditional entropy.** To evaluate the conditional entropy of the teachers' embeddings given the student's embedding, we need a kernel to learn their conditional distribution $\hat{p}(\mathsf{T}_k(\mathbf{X})|\mathsf{S}(\mathbf{X}))$ as presented in Figure 1. To this end, we use a parametric Gaussian model whose parameters $\mu_k(\mathsf{S}(\mathbf{X}))$ and $\Sigma_k(\mathsf{S}(\mathbf{X}))$ are learned during the student's training (Pichler et al., 2022).

**Loss function.** Following the above reasoning, we propose to train the student embedder $\mathsf{S}$ by minimizing the negative log-likelihood of the teachers' embeddings given the student's embedding, where the likelihood is estimated using Gaussian Kernels as follows:

$$
\begin{aligned}
\hat{\mathcal{L}}(\mathbf{X}, \mathsf{S}, \mathsf{T}_1, \ldots, \mathsf{T}_K) &= \frac{1}{K} \sum_{k=1}^{K} h(\mathsf{T}_k(\mathbf{X})|\mathsf{S}(\mathbf{X})) \\
&\leqslant \frac{1}{K} \sum_{k=1}^{K} \mathbb{E}_{\mathbf{X}} \Big[ -\log \mathcal{N}\big(\mathsf{T}_k(\mathbf{X}) \, \big| \, \mu_k(\mathsf{S}(\mathbf{X})), \Sigma_k(\mathsf{S}(\mathbf{X}))\big)\Big], \quad (3)
\end{aligned}
$$

where $\mathcal{N}(\cdot|\mu, \Sigma)$ is the Gaussian distribution with mean $\mu$ and covariance $\Sigma$. In our setting, minimizing the conditional entropy $h(\mathsf{T}_k(\mathbf{X})|\mathsf{S}(\mathbf{X}))$, exactly corresponds to maximizing the mutual information $I(\mathsf{T}_k(\mathbf{X}); \mathsf{S}(\mathbf{X})) = h(\mathsf{T}_k(\mathbf{X})) - h(\mathsf{T}_k(\mathbf{X})|\mathsf{S}(\mathbf{X}))$ since for each teacher $h(\mathsf{T}_k(\mathbf{X}))$ is constant w.r.t of the student. This also applies to the bound in Eq. 2.

**Training procedure.** We train both the student and the different kernels in an end-to-end fashion by minimizing the loss function $\hat{\mathcal{L}}$. It boils down to minimizing the negative log-likelihood of the teachers' embeddings given the student's embedding. We use the Adam optimizer to minimize the loss function. See Appendix E for the detailed training algorithm. To reduce the computational cost, we first embedded the entirety of the training set using the teachers and store them. We can then build training batches by sampling from the pre-computed embeddings.

**Baselines and Evaluation.** We consider two widely used multi-teacher feature distillation methods, MSE, used in SimReg (Navaneet et al., 2022) and Cosine similarity (see Appendix G for more information). To evaluate the representations learned by the student, for each modality, we run different benchmarks evaluating its performance on a wide variety of downstream tasks. For classification and regression tasks, we train a small feedforward network on top of the embeddings (the backbones are considered frozen) on different tasks and evaluate its performance.

## 4 Text Embedders

### 4.1 Experimental setting

We focus on distilling high-performing and large models into significantly smaller ones. Indeed, modern models in NLP are extremely large and costly to train[2]. Thus, we aim to produce the best possible models for a given weight category, pushing the size/performance of the Pareto frontier (Figure 2a), and not necessarily competing with the largest models. We distill from four teachers ranging from $433M$ parameters to $7B$ into students ranging from 20M to 335M parameters based on the nowflakes (Merrick et al., 2024) embedders.

**Teachers and student.** We select four freely available embedding models from the Huggingface hub (Wolf et al., 2020) (See Sec. C.1.2 for a detailed list of the teachers) whose evaluations are available in the MTEB benchmark (Muennighoff et al., 2023). To ensure having a point of comparison, we select teachers of different sizes and performances. Notably, SFR-Embeddings-R_2 is more than ten points stronger than the other three (smaller) teachers. As students we use snowflakes (Merrick, 2024; Merrick et al., 2024) models xs (22M), s (33M), m (109M) and l (335M) and we further train them using our distillation method (See Sec. C.1.4).

**Embedder evaluation.** Evaluating NLP models is notably challenging, and the common practice of evaluating a model using multi-task benchmarks may not be indicative of model capabilities (Liu et al.,

---

[2]`https://github.com/ills-montreal/nlp-distill`

Table 1: Performance of our distilled models compared to the strongest models of similar sizes from the MTEB Benchmark on classification tasks. Our 109M parameters model outperform significantly models 3 times bigger exhibiting exceptional information density.

| | | Task Model | Size | Amazon Counterfactual | Amazon Polarity | Amazon Reviews | Banking77 | Emotion | Imdb | MTOPDomain | MTOPIntent | Massive Intent | Massive Scenario | Toxic Conversations | Tweet Sentiment Extraction | Avg. |
|---|---|---|---|---|---|---|---|---|---|---|---|---|---|---|---|---|
| xs | Bas. | GIST | 23M | 72.9 | **87.2** | 42.6 | 84.2 | 52.1 | 78.5 | 94.8 | 77.7 | 73.2 | 76.7 | 72.9 | 59.9 | 72.7 |
| | | Ivysaur | 23M | 72.1 | 86.7 | 42.7 | 81.9 | 45.4 | 80.8 | 92.1 | 71.9 | 70.3 | 74.9 | 65.5 | 58.7 | 70.2 |
| | | gte-tiny | 23M | 71.8 | 86.6 | 42.6 | 81.7 | 44.7 | 80.5 | 91.8 | 69.9 | 70.1 | 74.9 | 71.0 | 58.6 | 70.3 |
| | MSE | Student-xs | 23M | 71.6 | 86.2 | 42.3 | 83.6 | 57.5 | **83.5** | 94.5 | 75.4 | 74.3 | **80.4** | 66.3 | 59.3 | 72.9 |
| | NLL | Student-xs | 23M | **76.5** | 84.9 | 42.4 | **85.8** | **58.0** | 81.1 | **95.2** | 79.9 | **75.8** | 80.4 | 68.1 | **60.1** | **74.0** |
| s | Bas. | bge-small-en-v1.5 | 33M | 73.8 | 92.8 | 47.0 | 85.7 | 47.8 | **90.6** | 93.4 | 74.8 | 74.8 | 78.7 | 69.9 | 60.5 | 74.1 |
| | | GIST | 33M | 75.3 | 93.2 | 49.7 | 86.7 | 55.9 | 89.5 | 95.5 | 79.1 | 75.5 | 79.2 | 72.8 | 61.0 | 76.1 |
| | | NoInstruct | 33M | 75.8 | **93.3** | **50.0** | 86.4 | 55.1 | 90.2 | 95.3 | 79.6 | 76.0 | 79.3 | 69.4 | **61.3** | 76.0 |
| | MSE | Student-s | 33M | 72.6 | 90.3 | 44.3 | 84.2 | 56.5 | 88.8 | 94.9 | 77.2 | 75.4 | **81.2** | 64.9 | 60.4 | 74.2 |
| | NLL | Student-s | 33M | **77.3** | 89.2 | 43.8 | **86.7** | **58.0** | 88.3 | **95.5** | **81.9** | **76.7** | 80.7 | 66.1 | 60.6 | 75.4 |
| m | Bas. | bge-base-en-v1.5 | 109M | 76.2 | 93.4 | 48.9 | 87.0 | 51.9 | **90.8** | 94.2 | 76.9 | 76.2 | 80.2 | 71.6 | 59.4 | 75.5 |
| | | GIST | 109M | 76.0 | **93.5** | **50.5** | 87.3 | 54.7 | 89.7 | 95.3 | 78.1 | 76.0 | 79.6 | 72.4 | 59.3 | 76.0 |
| | | e5-base-4k | 112M | 77.8 | 92.8 | 46.7 | 83.5 | 47.0 | 86.2 | 93.7 | 75.3 | 73.0 | 77.7 | 72.1 | 60.4 | 73.8 |
| | | e5-base-v2 | 110M | 77.8 | 92.8 | 46.7 | 83.5 | 47.0 | 86.2 | 93.7 | 75.3 | 73.0 | 77.7 | 72.1 | 60.4 | 73.8 |
| | MSE | Student-m | 109M | 76.6 | 89.1 | 44.7 | 87.2 | **60.8** | 88.0 | 95.7 | 81.6 | 77.7 | 82.2 | 67.3 | 60.5 | 76.0 |
| | NLL | Student-m | 109M | **79.6** | 89.5 | 45.8 | **88.0** | 59.7 | 88.3 | **96.2** | **83.9** | 78.6 | 82.7 | 67.1 | **61.3** | **76.7** |
| l | Bas. | bge-large-en-v1.5 | 335M | 75.8 | 92.4 | 48.2 | 87.8 | 51.5 | **92.8** | 94.6 | 79.5 | 77.6 | 80.5 | 70.9 | 59.9 | 76.0 |
| | | GIST | 335M | 75.6 | 93.4 | 49.1 | **88.1** | 54.7 | 91.2 | 95.2 | 78.2 | 76.2 | 79.3 | 71.9 | 59.2 | 76.0 |
| | | UAE-Large-V1 | 335M | 75.5 | 92.8 | 48.3 | 87.7 | 51.8 | 92.8 | 94.0 | 76.9 | 76.5 | 79.8 | 71.1 | 59.8 | 75.6 |
| | | ember-v1 | 335M | 76.1 | 92.0 | 47.9 | 87.9 | 52.0 | 92.8 | 94.6 | 75.3 | 77.4 | 80.5 | 71.4 | 60.0 | 76.0 |
| | | mxbai-embed-large-v1 | 335M | 75.0 | **93.8** | 49.2 | 87.8 | 50.9 | 92.8 | 94.0 | 76.8 | 76.2 | 80.0 | 71.5 | 59.7 | 75.6 |
| | MSE | Student-l | 335M | 77.3 | 84.5 | 43.4 | 86.0 | 60.0 | 82.7 | 95.1 | 79.8 | 76.3 | 81.3 | 65.8 | 60.2 | 74.4 |
| | NLL | Student-l | 335M | **81.5** | 88.1 | 45.9 | 86.9 | **60.4** | 88.2 | **95.6** | **83.2** | 77.5 | **81.4** | 67.7 | **62.2** | **76.5** |

2024). For lack of better options and because it is currently the most widely accepted benchmark, we rely on the evaluation provided by the MTEB benchmark (Muennighoff et al., 2023) on 33 tasks encompassing clustering (11 datasets), sentence similarity (10 datasets) and classification tasks (12 datasets). We compare our models with distilled and non-distilled ones from the MTEB leaderboard.

**Training set.** We gathered different common datasets used for training embedders and collected 6 million entries from the Huggingface Hub, including Specter (Cohan et al., 2020), T5 (Ni et al., 2021), Amazaon QA (McAuley & Leskovec, 2013), IMDB (Maas et al., 2011), SNLI (Bowman et al., 2015), QQP triplets from Quora, AG News (Zhang et al., 2015), MEDI dataset (Su et al., 2023) and the DAIL Emotion dataset (Saravia et al., 2018). We provide the dataset statistics in Sec. C.1.1. The datasets are all flattened, such that if the original had two columns (e.g., sentence 1 and 2 in the SNLI dataset), we end up with twice the number of entries, one for each sentence, and we deduplicated the dataset. Models are trained for two epochs with batch size 16 on NVIDIA V100.

## 4.2 Distillation performance

**Task performance.** Our method produces models that exhibit strong performance on a large variety of tasks, ranking first amongst all models of similar size in the MTEB benchmark on most of the tasks (Figure 2b). Notably, we observe that our method produces models that are competitive for almost all the tasks, whereas other models appear more specialized. We provide the actual accuracy of our models on classification tasks in Tab. 1. We provide the full results for all model sizes in Sec. C.2.1.

**Pareto frontier.** Our goal with distillation is to increase information density of models to reduce computational costs and memory footprint, we show in Figure 2a that our method can pack more information into fixed-size models. Interestingly, our medium-sized model (109M parameters) outperforms all the models three times its size and even our 335M model under the same training setting. In addition, our small models outperform all previous model of their weight category, notably yielding a 2-point gain on average classification accuracy on the MTEB over the previous *state-of-the-art* efficient GIST-based embedders (Solatorio, 2024).

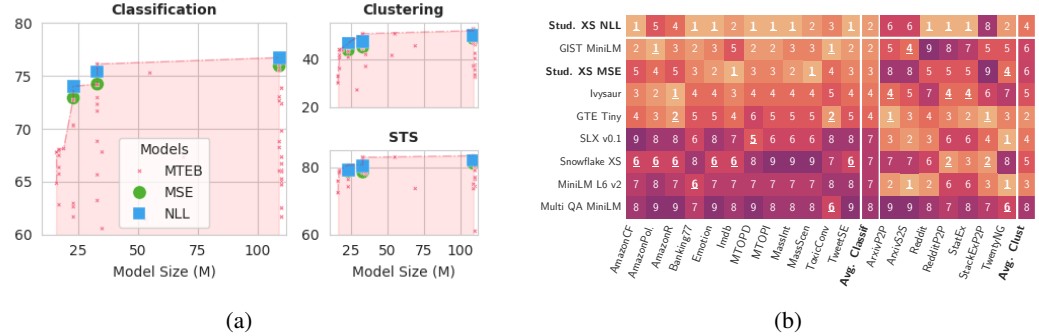

(a)                        (b)

Figure 2: (a) **Pareto frontier size/performance in NLP.** Our method (in blue) yields Pareto optimal model. (b) **Global ranking of embedders** on clustering and classification tasks for our xs model (23M). The NLL-distilled model rank 1 in most tasks and in average, outperforming all other baselines of its weight category and closing the gap with models 10 times bigger.

**Comparison with standard MSE distillation.** Consistent with results from reinforcement learning and interval estimation(Stewart et al., 2023), training the student to match the teachers' embeddings using MSE loss results in consistently worse models.

**Limitations of the embedding space structure.** Our metric, which optimizes mutual information between the student and teachers, does not impose structure on the embedding space. Given that information remains invariant under invertible transformations, let $f_1$ and $f_2$ be differentiable and invertible mapping functions (diffeomorphisms); thus, $I(X;Y) = I(f_1(X); f_2(Y))$. Consequently, our objective does not ensure the preservation of structural properties, such as pairwise cosine similarity, in the teachers' embedding space. Nonetheless our method maintains competitive performance in both clustering and Semantic Textual Similarity (STS) (see Appendix C.2).

## 5 Molecular Embedders

We further our method in molecular modeling, enabling the distillation of a student with models leveraging different modalities to represent a molecule: text, graph, and 3D point clouds.

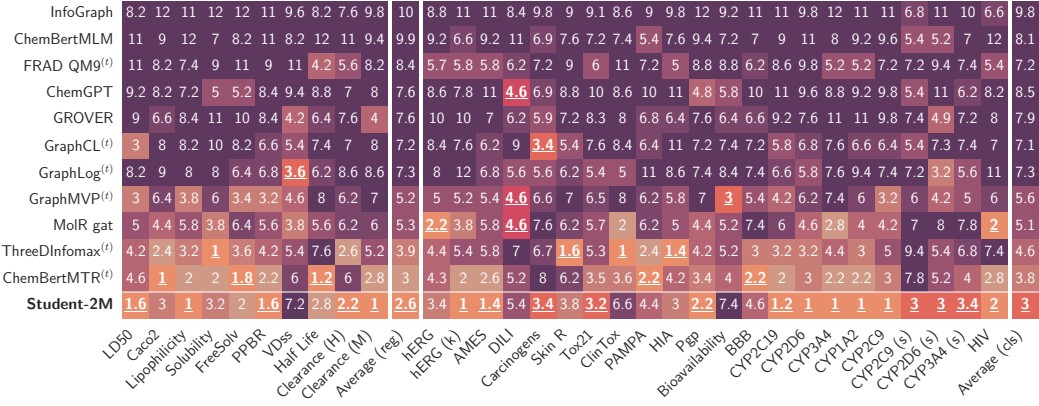

Figure 3: **Ranking on the TDC ADMET tasks.** Our student consistently achieves competitive performances across the evaluated tasks compared to its teachers (denoted by $^{(t)}$) and the other baselines, achieving the best average rank for both regression and classification tasks.

### 5.1 Experimental setting

**Teachers and architecture.** We use eight teachers trained on different modalities: SMILES (textual representation of the molecular graph) (Ahmad et al., 2022), 2D molecular graphs (You et al., 2020; Xu et al., 2021; Liu et al., 2022; Stärk et al., 2021), and 3D structures (Feng et al., 2023). We identify

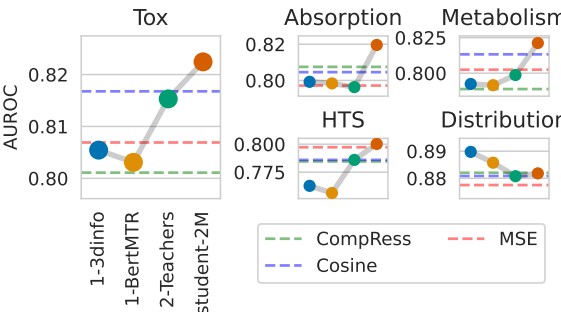

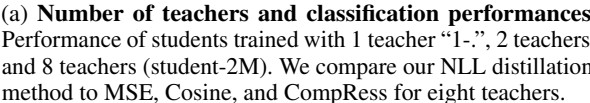

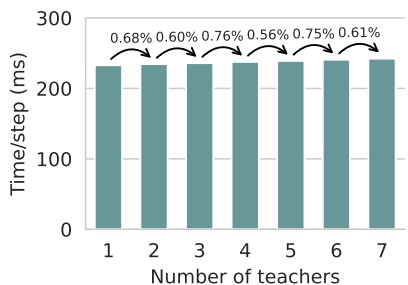

(a) **Number of teachers and classification performances.** Performance of students trained with 1 teacher "1-.", 2 teachers, and 8 teachers (student-2M). We compare our NLL distillation method to MSE, Cosine, and CompRess for eight teachers.

(b) **Computational overhead.** Evolution of runtime for a training step as a function of the number of teachers. The computational overhead induced by an additional teacher represents less than 1% of the total runtime on a batch.

the teachers with $^{(t)}$ such as ChemBERTaMTR$^{(t)}$, and use a 2D-GNN (Graph Isomorphism Network: GIN (Hu et al., 2020)) for our student (for more details see Sec. B.1)[3].

**Evaluation setting.** We evaluated all models on the ADMET (Absorption, Distribution, Metabolism, Excretion, Toxicity) tasks of the Therapeutic Data Commons platform (TDC) (Huang et al., 2021) and on a high-throughput screening task (HTS), (HIV (Wu et al., 2018)). We record the test performance over five runs (details on the evaluation procedure in Sec. B.3). We trained our models on MOSES, a processed version of the ZINC Clean Leads dataset (Polykovskiy et al., 2018), containing 2 million samples, and on ZINC-250k (Irwin & Shoichet, 2005), consisting of 250,000 samples. The performances of the model trained on 250k samples can be found in Sec. B.1. Both are public datasets of commercially available compounds designed to be used in various therapeutic projects.

## 5.2 Results

**Overall performance.** We compare the performance of the student model with the teachers and other baseline embedders on the different tasks. The results (average rank) for each task are presented in Figure 3. Our student model achieves the best performance on both the regression and classification tasks, delivering the most accurate predictions across a majority of tasks. This suggests that our method generates informative representations, providing high-quality molecular descriptors.

**Single teacher vs. multi-teachers.** To assess the impact of training a student with multiple teachers, we trained students to distill the knowledge of a single teacher and two teachers, and compared the results to those of our student trained with eight teachers. We selected two of the best-performing baselines as teachers: ChemBERTaMTR-77M (Ahmad et al., 2022) and 3D-infomax (Stärk et al., 2021). We then trained student models on the 2M-molecules dataset. Figure 4a displays the performances of each of these student models on the regression tasks. Training with multiple teachers consistently outperforms training with a single teacher, except on the Blood-Brain Barrier (BBB) task (the only Distribution classification task), which is also one of the tasks our model struggles the most with. For the BBB benchmark, we noticed it is one of the datasets where all results are among the most tightly packed (variations within 1.45 times the average standard deviation of the results), and whose data distribution differs the most from the training set, which could explain the slightly lower average performance of the 8-teacher student compared to the 1 or 2-teacher students. Overall, using multiple teachers significantly improves performance, with the best performance achieved when training with all eight teachers (additional results are available in Sec. B.4).

**Comparison to baselines.** Figure 4a also compares the performance of our NLL distillation method to MSE, cosine, and CompRess distillation for eight teachers. Overall, in the evaluation of classification tasks, our NLL distillation method outperformed the Cosine and MSE distillation methods. This observation goes beyond the results of classification tasks, as we also observed that the NLL distillation method consistently outperforms the other two methods on all evaluated task categories (see Sec. B.1.3 for more details).

---

[3]`https://github.com/ills-montreal/mol-distill`

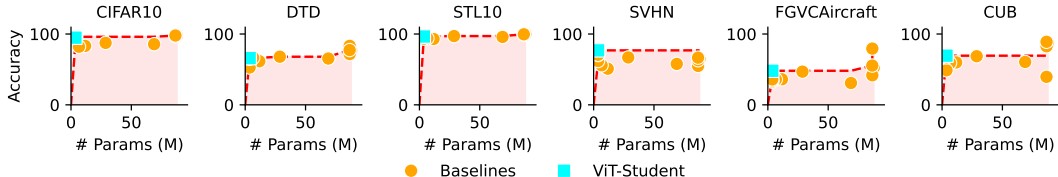

Figure 5: **Pareto frontier of vision models.** The figure compares the performance of student model distilled using our method (named ViT-Student shown with color blue) with baselines (shown in yellow) across various datasets. The distilled student consistently lies on the Pareto frontier.

**Computational complexity.** Training our molecular embedders on the largest dataset (2 M molecules) takes approximately 50 hours on 6 A6000 GPUs. We evaluated the computational overhead induced by the multi-teacher setting in Figure 4b. The runtime of a training step increases linearly with the number of teachers: $+1.57ms$ per teacher, representing less than $1\%$ of the total runtime.

## 6 Image Embedders

For our final modality, vision, we aim to assess whether our method can deliver competitive performance compared to other baseline models (teachers, and MSE, Cosine, and CompRess student), especially on fine-grained vision classification tasks. In the following subsections, we outline the experimental setup used to investigate these questions and present the results. Additional details, including hyperparameter tuning and the augmentations applied, can be found in Sec. D.3.

### 6.1 Experimental setting

**Teachers and evaluations.** Given the increasing use of Vision Transformers, we used large transformer models (Swin (Liu et al., 2021b), DINOv2 (Oquab et al., 2023), ViT (Dosovitskiy et al., 2021), and BEiT (Bao et al., 2022), with around 87 million parameters) as teachers, and selected a smaller Vision Transformer, PVTv2 (Wang et al., 2022c), with 3.7 million parameters, as the student. We also use some CNN based modes with different sizes as baselines to have a more comprehensive comparison of our student's representation abilities (refer to Sec. D.1 for more details).

**Training set.** We include fine-grained datasets such as DTD (Cimpoi et al., 2014), FGVCAircraft (Maji et al., 2013), and CUB (Welinder et al., 2010), alongside CIFAR10 (Krizhevsky et al., 2009), SVHN (Netzer et al., 2011), STL10 (Coates et al., 2011) for the vision experiment. These allows us to assess the performance of our approach on a variety of challenging and detailed classification tasks. Refer to Sec. D.2 for details of the datasets meta-data [4].

### 6.2 Results on Vision Transformer

To further evaluate our method, we conducted experiments using Vision Transformer (ViT) teachers. As shown in Figure 5, the distilled student model trained with our approach consistently lies on the Pareto frontier, for each task, showing a superior trade-off between accuracy and model size. Notably, our distilled student achieves the best performance among other distillation methods and other baseline models within its respective size categories, with results comparable to large ViT teachers ($20\times$ more parameters). This demonstrates our method's ability to effectively transfer knowledge from large, complex teacher models to smaller, more efficient student models, while maintaining comparable performance. Additional results in Sec. D.4 show that our method generalizes well to unseen vision datasets, improving other distillation baselines, and effectively integrates diverse task-specific teachers without performance conflicts, confirming its robustness across domains.

---

[4] https://github.com/ills-montreal/vision-distill/

# 7   Limitations

Our method focuses on training student embedding models for diverse, unknown tasks; for single, pre-defined tasks, task-specific distillation may be more effective. As with any distillation approach—especially multi-teacher distillation—there is an overhead, either computational (if teacher embeddings are generated on-the-fly) or memory-intensive (if precomputed). We mitigate this by precomputing and storing embeddings, requiring approximately 100GB of disk space for our largest text-based teacher. The quality of our student embeddings depends on the relevance of the teachers to the downstream tasks. While task-specific teachers provide limited benefits outside their domain, they do not degrade performance when combined with task-relevant teachers (Sec. D.4). Our optimization metric maximizes mutual information between student and teachers but does not explicitly structure the embedding space, potentially limiting performance in tasks like clustering. For textual embeddings, we observe significant gains in classification (where embeddings train a small classifier) but more modest improvements in clustering and STS tasks, which rely on embedding dot products for similarity assessment (Sec. C.2.2).

# 8   Conclusions and Future Work

We proposed a theoretically grounded task-agnostic distillation mechanism that leverages interval estimation through Gaussian kernels in high dimensions to distill a more informative representation from multiple teachers to a single student. We demonstrated that our objective serves as a proxy for maximizing the mutual information and reconstructive capacity of the student model in relation to the teachers. We experimentally validated that our method is more efficient than point estimation-based multi-teacher feature distillation methods such as MSE or cosine-based distillation mechanisms. We demonstrated the superior performance of our method compared to others across three different modalities and numerous downstream tasks. In future work, we aim to extend this distillation approach to cross-modal distillation, enhancing the model's capabilities by leveraging task-agnostic cross-modal information.

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

# Appendix

## Table of Contents

# A Proofs

We denote $\mathbf{X}$ as the random variable over $\mathcal{X}$ that describes the input distribution. We suppose we have access to a dataset $\mathcal{D} = \{\mathbf{x}_i\}_{i=1}^n \subset \mathcal{X}$ of inputs drawn following $p_{\mathbf{X}}$ and different embedders $\mathsf{T}_k : \mathcal{X} \to \mathbb{R}^{d_k}$, $k \in \{1, \ldots, K\}$, that map the inputs to different embedding spaces. We denote $\mathbf{Z_k} = \mathsf{T}_k(\mathbf{X})$ as the random variable over $\mathbb{R}^{d_k}$ that describes the embedding of the input distribution in the $k$-th embedding space and by $\mathbf{U} = \mathsf{S}(\mathbf{X})$ the random variable over $\mathbb{R}^d$ that describe the embedding of the input distribution in the student embedding space. We denote by $\mathbf{z}_i^k = \mathsf{T}_k(\mathbf{x}_i)$ the embedding of $\mathbf{x}_i$ in the $k$-th embedding space. We are interested in learning a representation that captures the information contained in all the embeddings.

Let us consider a task characterized by a target set $\mathcal{Y}$ of discrete concepts and the feature space $\mathcal{X}$ with joint probability measure $P_{\mathbf{YX}} \in \mathcal{P}(\mathcal{Y} \times \mathcal{X})$. For every projection of the features through the different teachers, the Bayes decision rule $c_{\mathsf{T}_k}^* \triangleq \arg\max_{c:\mathbb{R}^{d_k} \to \mathcal{Y}} \mathbb{E}_{\mathbf{XY}}\big[\mathbb{1}\left[c(\mathsf{T}_k(\mathbf{X})) = \mathbf{Y}\right]\big]$ and similarly for the student: $c_{\mathsf{S}}^* \triangleq \arg\max_{c:\mathbb{R}^d \to \mathcal{Y}} \mathbb{E}_{\mathbf{XY}}\big[\mathbb{1}[c(\mathsf{S}(\mathbf{X})) = \mathbf{Y}]\big]$.

We leverage the following recent result from (Darrin et al., 2024):

**Proposition A.1.** *Let $C_{\mathsf{T}_k} = c_{\mathsf{T}_k}^*(\mathsf{T}_k(\mathbf{X}))$ and $C_{\mathsf{S}} = c_{\mathsf{S}}^*(\mathsf{S}(\mathbf{X}))$ denote the outcome of the Bayes classifier observing the output of the teacher $\mathsf{T}_k$ and the student $\mathsf{S}$, respectively*

$$\Pr\left(C_{\mathsf{S}} \neq C_{\mathsf{T}_k}\right) \leqslant 1 - \exp\big(-h(\mathsf{T}_k(\mathbf{X})|\mathsf{S}(\mathbf{X}))\big). \tag{4}$$

## A.1 Proof of Theorem 3.2

By applying the above proposition to all the terms in Eq. 1, we obtain the following bound on the loss function:

**Proposition 1** (Upper bound)**.**

$$\mathcal{L}^*(\mathbf{XY}, \mathsf{S}, \mathsf{T}_1, \ldots, \mathsf{T}_K) \leqslant \frac{1}{K} \sum_{k=1}^K \left(1 - \exp\big(-h(\mathsf{T}_k(\mathbf{X})|\mathsf{S}(\mathbf{X}))\big)\right) \tag{5}$$

$$\leqslant 1 - \exp\left(-\underbrace{\frac{1}{K} \sum_{k=1}^K h(\mathsf{T}_k(\mathbf{X})|\mathsf{S}(\mathbf{X}))}_{\textit{Negative log likelihood}}\right). \tag{6}$$

*Proof.*

$$\mathcal{L}^*(\mathbf{XY}, \mathsf{S}, \mathsf{T}_1, \ldots, \mathsf{T}_K) \leqslant \frac{1}{K} \sum_{k=1}^K \left(1 - \exp\big(-h(\mathsf{T}_k(\mathbf{X})|\mathsf{S}(\mathbf{X}))\big)\right)$$

$$\leqslant 1 - \frac{1}{K} \sum_{k=1}^K \exp\big(-h(\mathsf{T}_k(\mathbf{X})|\mathsf{S}(\mathbf{X}))\big)$$

$$\leqslant 1 + \frac{1}{K} \sum_{k=1}^K -\exp\big(-h(\mathsf{T}_k(\mathbf{X})|\mathsf{S}(\mathbf{X}))\big)$$

$$\leqslant 1 - \exp\left(-\frac{1}{K} \sum_{k=1}^K h(\mathsf{T}_k(\mathbf{X})|\mathsf{S}(\mathbf{X}))\right).$$

We simply rearrange the terms and use the fact that $x \mapsto -\exp(-x)$ is concave to interchange the sum and the exponential. $\qquad\square$

# B Molecular Modelling

## B.1 Model architecture

We trained a 10-layer GINE (Hu et al., 2020) neural network with a 512 hidden dimension, using a 2-layer network for the message passing process. We use the atomic number of each node as input, as well as possible chirality information, and the nature of the bond between each pair of nodes. We use a batch size of 256 and a learning rate of $1e-4$ to train the model for 400 epochs on the 250k dataset and 200 epochs on the 2M dataset. For the teacher-specific kernels, we used a 3-layer MLP with a hidden size of 1024.

### B.1.1 Chosen Teachers

The teachers used to train our molecular modeling students are summed up in Tab. 2. We gathered various representation models for molecular modeling, with different pre-training objectives, input modalities, architectures, and training datasets.

Table 2: Description of all teachers used in our experiments.

| Model name | SMILES | 2D-GNN | 3D-GNN | Architecture | Out size | Dataset (size) |
|---|---|---|---|---|---|---|
| GraphCL(You et al., 2020) | | ✓ | | GIN | 300 | GEOM (Axelrod & Gómez-Bombarelli, 2022) (50k) |
| GraphLog(Xu et al., 2021) | | ✓ | | GIN | 300 | GEOM (Axelrod & Gómez-Bombarelli, 2022) (50k) |
| GraphMVP(Liu et al., 2022)[1] | | ✓ | | GIN | 300 | GEOM (Axelrod & Gómez-Bombarelli, 2022) (50k) |
| 3D-infomax(Stärk et al., 2021)[1] | | ✓ | | PNA | 800 | QMugs (Isert et al., 2021) (620k) |
| ChemBERT MTR(Ahmad et al., 2022)[2] | ✓ | | | RoBERTa | 384 | PubChem (Kim et al., 2022) (5M, 10M, 77M) |
| 3D-fractional(Feng et al., 2023) | | | ✓ | TorchMD-net | 256 | PCQM4Mv2(Hu et al., 2021) (3.7M) |

### B.1.2 Architecture influence

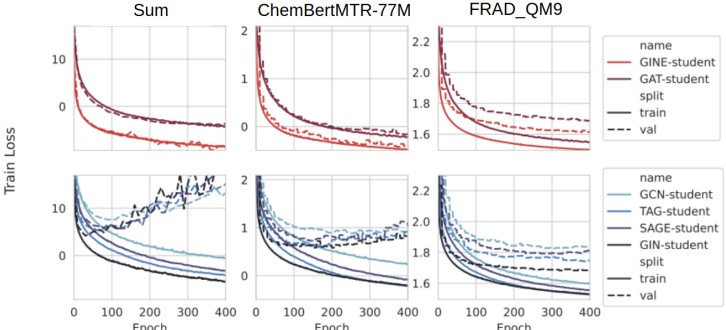

Figure 6: Training loss of different students using different GNN architectures on the ZINC-250k dataset.

Figure 6 shows the training loss of the student model with different GNN architectures on the ZINC-250k dataset. In particular, we compared the GINE architecture with a Graph Convolutional Network (GCN) (Morris et al., 2021), a Graph Attention Network (GAT) (Brody et al., 2022), a GraphSAGE (SAGE) (Hamilton et al., 2018), a Toplogy Adaptative Graph Convolutional Network (TAG) (Brody et al., 2022), and a GIN Network, that separates from the GINE architecture by the fact that it does not take edge features into account (Xu et al., 2019). We observe that the GINE architectures outperform the other architectures, with a lower training loss, a faster convergence, and a lower validation loss. The Graph attention network (GAT) is the second best performing architecture, but it is still outperformed by the GINE architecture. These two architectures are the only ones to use the edge embeddings in the message passing process, which could explain their better performance.

---

[1]Models aiming at incorporating 3D information into 2D-GNNs models.
[2]We used the three versions of ChemBERT-MTR models trained on 5M, 10M, and 77M.

Indeed, all other architectures perform worse, especially when considering their validation loss computed on 10% of the training set. Specifically, the GIN architecture, not using edge feature, performs significantly worse than the GINE architecture, while having a similar architecture.

For our experiments, we decided to use the GINE architecture, as it performs the best during training and converges faster than the other architectures.

### B.1.3 Additional results on the TDC datasets

Table 3: Average rank of each model on the ADMET and HTS downstream tasks from the TDC (Huang et al., 2021) platform. Our student outperforms all baselines, including teachers, on average.

|  | Absorption | Distribution | Metabolism | Excretion | Tox | HTS | Avg |
|---|---|---|---|---|---|---|---|
| InfoGraph | 13.50 | 13.27 | 13.32 | 11.40 | 11.98 | 9.40 | 12.14 |
| ChemBertMLM-10M | 10.65 | 11.00 | 10.70 | 13.80 | 11.11 | 14.60 | 11.98 |
| FRAD QM9$^{(t)}$ | 10.57 | 11.13 | 10.38 | 8.33 | 10.04 | 7.80 | 9.71 |
| ChemGPT-1.2B | 9.55 | 11.73 | 11.75 | 10.73 | 10.86 | 11.20 | 10.97 |
| GROVER | 10.43 | 8.33 | 11.25 | 8.53 | 10.38 | 11.00 | 9.99 |
| GraphCL$^{(t)}$ | 10.89 | 8.53 | 9.45 | 10.13 | 8.70 | 9.80 | 9.58 |
| GraphLog$^{(t)}$ | 11.05 | 7.80 | 9.07 | 10.53 | 8.93 | 14.00 | 10.23 |
| GraphMVP$^{(t)}$ | 7.20 | 6.20 | 7.85 | 9.80 | 7.49 | 8.80 | 7.89 |
| MolR gat | 6.95 | 7.60 | 8.30 | 8.53 | 6.49 | _3.40_ | 6.88 |
| ThreeDInfomax$^{(t)}$ | _4.17_ | _6.00_ | 7.58 | 7.13 | 6.16 | 10.40 | 6.91 |
| ChemBertMTR-77M$^{(t)}$ | **3.50** | **4.27** | 5.75 | 5.00 | 6.03 | 4.20 | _4.79_ |
| MSE | 8.07 | 6.40 | 5.55 | 6.33 | 7.55 | **3.00** | 6.15 |
| Cosine | 5.51 | 6.13 | _3.60_ | _4.33_ | **4.97** | 6.20 | 5.13 |
| student-250k | **3.55** | 6.20 | **2.70** | **2.40** | 4.99 | 3.80 | **3.94** |
| student-2M | 4.40 | **5.40** | **2.75** | **3.00** | _4.34_ | **2.40** | _3.72_ |

The average rank of each model in each task category can be found in Tab. 3. Surprisingly, the performances of the "student-250k" and "student-2M" models are similar on average. Specifically, the student-250k model outperforms the student-2M model on regression datasets notably, by achieving the best performances on the FreeSolv (Mobley & Guthrie, 2014) and Lipophilicity (Wenlock & Tomkinson, 2021) tasks. This suggests that our method can leverage the diversity of the teachers to learn more informative representations, even when trained on a smaller dataset of 250k datapoints.

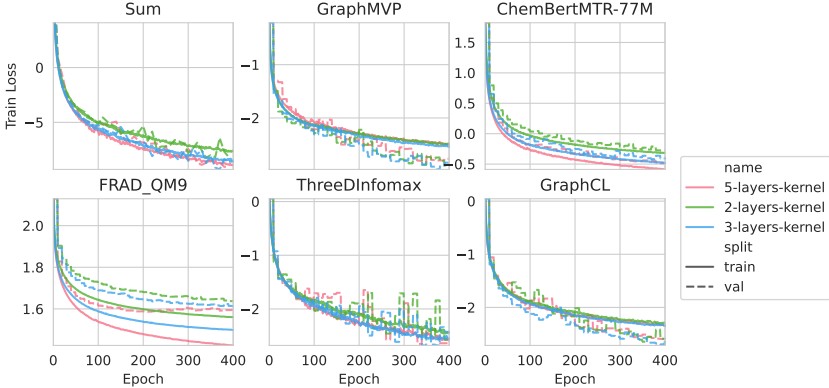

Figure 7: Training loss of the student model along the training with different kernel-size on the ZINC-250k dataset.

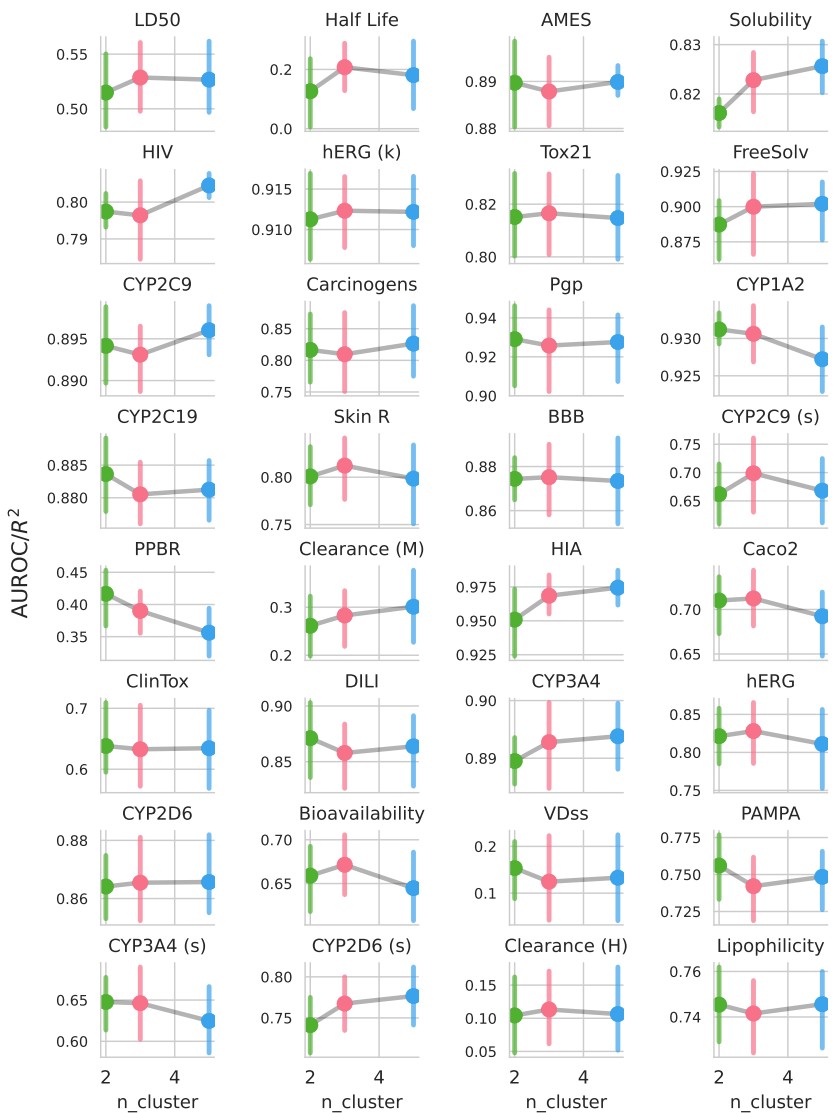

Figure 8: Test AUROC/$R^2$ score of the students on the classification/regression tasks, trained with different kernel-size on the ZINC-250k dataset.

## B.2 Kernel's predictive power

Our method relies on teacher-specific heads to distill the knowledge of each teacher. In this section, we wish to evaluate the impact of the choice of these kernels and their predictive power (in terms of depth) on the performance and training of the student model.

We performed this experiment with kernels of depth 2, 3, and 5, and we trained the student model with these kernels on the ZINC-250k dataset and evaluated the performance of the student model on the ADMET and HTS downstream tasks.

First, during the training, as expected, the more powerful the kernel, the lower the training loss is (see Figure 7), even though the difference is significant, especially between the students using kernels of depth 3 and 5. Overall, the performances of each student on the downstream tasks are similar, underlining the robustness of our method regarding the choice of the kernel's depth (see Figure 8). For our experiments in the main paper, we used a kernel of depth 3, as it enables the best trade-off

between computational complexity, and training convergence while providing competitive results on the downstream tasks.

## B.3 Evaluation details

### B.3.1 Benchmark Choice

We selected a total of 32 tasks, extracted from the Therapeutic Data Commons (Huang et al., 2021) platform, 8 absorption tasks, 3 distribution tasks, 8 metabolism tasks, 3 excretion tasks, 9 toxicity tasks and 1 high-throughput screening task. A summary of the tasks considered can be found in Tab. 4, with their corresponding size (total number of samples) and type (classification or regression). For all tasks, we computed 5 conformations for each molecule, and used the least energetic as an input of our 3D models.

Table 4: Tasks extracted from the Therapeutic Data Commons platform considered in our experiments.

| Category | Model | Task | cls | reg |
|---|---|---|---|---|
| Absorption | P-glycoprotein Inhibition | 1212 | ✓ | |
| | AqSolDB | 9982 | | ✓ |
| | Lipophilicity | 4200 | | ✓ |
| | Caco-2 Permeability | 906 | | ✓ |
| | Human Intestinal Absorption | 578 | ✓ | |
| | FreeSolv | 642 | | ✓ |
| | PAMPA Permeability | 2035 | ✓ | |
| | Oral Bioavailability | 640 | ✓ | |
| Distribution | Plasma-Protein BDR | 1614 | | ✓ |
| | Blood-Brain barrier | 1975 | ✓ | |
| | VDss | 1130 | | ✓ |
| Metabolism | CYPP450 3A4 Inhib. | 12328 | ✓ | |
| | CYPP450 1A2 Inhib. | 12579 | ✓ | |
| | CYPP450 2C19 Inhib. | 12665 | ✓ | |
| | CYPP450 2C9 Inhib. | 12092 | ✓ | |
| | CYPP450 2D6 Inhib. | 13130 | ✓ | |
| | CYPP450 2D6 Substrate | 664 | ✓ | |
| | CYPP450 3A4 Substrate | 667 | ✓ | |
| | CYPP450 2C9 Substrate | 666 | ✓ | |
| Excretion | Clearance hepatocyte | 1020 | | ✓ |
| | Half Life | 667 | | ✓ |
| | Clearance microsome | 1102 | | ✓ |
| Toxicity | Tox21 | 7831 | ✓ | |
| | hERG | 13445 | ✓ | |
| | | 648 | ✓ | |
| | Acute Toxicity LD50 | 7385 | | ✓ |
| | Ames Mutagenicity | 7255 | ✓ | |
| | ClinTox | 1484 | ✓ | |
| | Carcinogens | 278 | ✓ | |
| | Drug Induced Liver Injury | 475 | ✓ | |
| | Skin Reaction | 404 | ✓ | |
| HTS | HIV | 40000 | ✓ | |

### B.3.2 Evaluation Procedure

For every task, we opted for a random split since we obtained similar results to a scaffold split, with a faster computation time, with a ratio of 70/10/20 for the train/validation/test sets. For all tasks, we compute the embeddings generated by each model on the task. We then train a 2 layer perceptron with a hidden size of 128 on the task for $\min(100, 200 * \frac{5000}{\text{task size}})$ epochs (to limit the compute time on large tasks) with a learning rate of $1e - 3$. We then select the best checkpoint according to the validation performances and report the test metrics of this checkpoint.

### B.3.3 Evaluation Metrics

We repeat this process five times with different seeds in the train-val-test splits in order to enable the establishment of robust rankings using autorank (Herbold, 2020). We decided to report the ranks of the models to enable the comparison of the models on both classification and regression by simply averaging the rank. To compute the rank on all tasks, we rely on the AUROC score for classification tasks and the $R^2$ score for regression tasks. For the excretion tasks, since the regression labels have a large variance, we decided to apply the regression on the log-values and report the $R^2$ score on the log-values.

## B.4 Single-Teacher setting

To assess the impact of the multi-teacher setting on the performance of the student model, we trained students to distill the knowledge of a single teacher. We used only the two best performing teachers, 3D-infomax (Stärk et al., 2021) and ChemBERTaMTR (Ahmad et al., 2022), to train the student model on the 2M datapoints dataset. We also train a student with both teachers, to see if those two teachers are sufficient to achieve the same performance as the models we presented in the core of the paper.

Figure 9 shows how these students underperform compared to a student trained with all teachers, in terms of AUROC for classification tasks and $R^2$ for regression tasks respectively. These tables also show that the student trained with both teachers performs better than each student trained with only one teacher. All results are aggregated in Tab. 6 and Tab. 5.

Table 5: Performance of the student models trained with only the best teacher ("1-ChemBertMTR"), the second-best teacher ("1-3dinfo"), both teachers together ("2-teachers"), and "student-2M" on regression tasks (R2).

| | avg | Absorption Caco2 | Absorption FreeSolv | Absorption Lipophilicity | Absorption Solubility | Tox LD50 |
|---|---|---|---|---|---|---|
| 1-3dinfo | $0.392\pm_{0.317}$ | $0.654\pm_{0.041}$ | $0.822\pm_{0.044}$ | $0.583\pm_{0.047}$ | $0.798\pm_{0.010}$ | $0.471\pm_{0.048}$ |
| 1-BertMTR | $0.405\pm_{0.309}$ | $0.660\pm_{0.026}$ | $0.829\pm_{0.031}$ | $0.582\pm_{0.044}$ | $0.803\pm_{0.010}$ | $0.480\pm_{0.023}$ |
| 2-Teachers | $\mathbf{0.449}\pm_{\mathbf{0.312}}$ | $\mathbf{0.692}\pm_{\mathbf{0.043}}$ | $\mathbf{0.882}\pm_{\mathbf{0.034}}$ | $\mathbf{0.688}\pm_{\mathbf{0.028}}$ | $0.812\pm_{0.012}$ | $\mathbf{0.497}\pm_{\mathbf{0.033}}$ |
| student-2M | $\underline{\mathbf{0.476}}\pm_{\mathbf{0.301}}$ | $\mathbf{0.687}\pm_{0.045}$ | $0.878\pm_{0.036}$ | $\underline{\mathbf{0.739}}\pm_{\mathbf{0.021}}$ | $\underline{\mathbf{0.822}}\pm_{\mathbf{0.005}}$ | $\underline{\mathbf{0.543}}\pm_{\mathbf{0.041}}$ |

| | Distribution PPBR | Distribution VDss | Excretion Clearance (H) | Excretion Clearance (M) | Excretion Half Life |
|---|---|---|---|---|---|
| 1-3dinfo | $0.316\pm_{0.062}$ | $0.130\pm_{0.146}$ | $0.048\pm_{0.095}$ | $0.137\pm_{0.083}$ | $-0.037\pm_{0.254}$ |
| 1-BertMTR | $0.347\pm_{0.070}$ | $\mathbf{0.145}\pm_{\mathbf{0.072}}$ | $0.051\pm_{0.148}$ | $0.136\pm_{0.110}$ | $0.017\pm_{0.195}$ |
| 2-Teachers | $\mathbf{0.419}\pm_{\mathbf{0.032}}$ | $\mathbf{0.172}\pm_{\mathbf{0.098}}$ | $\mathbf{0.066}\pm_{\mathbf{0.075}}$ | $\mathbf{0.199}\pm_{\mathbf{0.075}}$ | $\mathbf{0.061}\pm_{\mathbf{0.135}}$ |
| student-2M | $\mathbf{0.389}\pm_{\mathbf{0.050}}$ | $0.138\pm_{0.115}$ | $\underline{\mathbf{0.069}}\pm_{\mathbf{0.060}}$ | $\underline{\mathbf{0.348}}\pm_{\mathbf{0.062}}$ | $\underline{\mathbf{0.144}}\pm_{\mathbf{0.205}}$ |

Table 6: Performance of the student models trained with only the best teacher ("1-ChemBertMTR"), the second-best teacher ("1-3dinfo"), both teachers together ("2-teachers"), and "student-2M" on classification tasks (AUROC).

| | avg | Absorption Bioavailability | Absorption HIA | Absorption PAMPA | Absorption Pgp | Distribution BBB | HTS HIV |
|---|---|---|---|---|---|---|---|
| 1-BertMTR | $0.801\pm_{0.101}$ | $0.631\pm_{0.059}$ | $\mathbf{0.910}\pm_{\mathbf{0.054}}$ | $0.719\pm_{0.020}$ | $\mathbf{0.933}\pm_{\mathbf{0.021}}$ | $\mathbf{0.886}\pm_{\mathbf{0.022}}$ | $0.756\pm_{0.005}$ |
| 1-3dinfo | $0.803\pm_{0.100}$ | $0.631\pm_{0.050}$ | $0.899\pm_{0.056}$ | $\mathbf{0.737}\pm_{\mathbf{0.021}}$ | $0.930\pm_{0.024}$ | $\mathbf{0.890}\pm_{\mathbf{0.024}}$ | $0.763\pm_{0.010}$ |
| 2-Teachers | $\mathbf{0.808}\pm_{\mathbf{0.097}}$ | $\mathbf{0.641}\pm_{\mathbf{0.055}}$ | $0.868\pm_{0.051}$ | $\underline{\mathbf{0.749}}\pm_{\mathbf{0.009}}$ | $0.927\pm_{0.026}$ | $0.881\pm_{0.026}$ | $\mathbf{0.786}\pm_{\mathbf{0.012}}$ |
| student-2M | $\underline{\mathbf{0.825}}\pm_{\mathbf{0.096}}$ | $\underline{\mathbf{0.653}}\pm_{\mathbf{0.055}}$ | $\underline{\mathbf{0.959}}\pm_{\mathbf{0.026}}$ | $0.730\pm_{0.024}$ | $\underline{\mathbf{0.936}}\pm_{\mathbf{0.024}}$ | $0.882\pm_{0.020}$ | $\underline{\mathbf{0.800}}\pm_{\mathbf{0.014}}$ |

| | Metabolism CYP1A2 | Metabolism CYP2C19 | Metabolism CYP2C9 (s) | Metabolism CYP2C9 | Metabolism CYP2D6 (s) | Metabolism CYP2D6 | Metabolism CYP3A4 (s) | Metabolism CYP3A4 |
|---|---|---|---|---|---|---|---|---|
| 1-BertMTR | $0.916\pm_{0.008}$ | $0.866\pm_{0.007}$ | $0.622\pm_{0.088}$ | $0.874\pm_{0.006}$ | $\mathbf{0.729}\pm_{\mathbf{0.021}}$ | $0.839\pm_{0.010}$ | $\mathbf{0.638}\pm_{\mathbf{0.017}}$ | $0.848\pm_{0.014}$ |
| 1-3dinfo | $0.916\pm_{0.005}$ | $0.870\pm_{0.006}$ | $\mathbf{0.630}\pm_{\mathbf{0.069}}$ | $0.874\pm_{0.005}$ | $0.721\pm_{0.028}$ | $0.836\pm_{0.008}$ | $\mathbf{0.638}\pm_{\mathbf{0.015}}$ | $0.855\pm_{0.007}$ |
| 2-Teachers | $\mathbf{0.924}\pm_{\mathbf{0.006}}$ | $\mathbf{0.871}\pm_{\mathbf{0.009}}$ | $0.627\pm_{0.086}$ | $\mathbf{0.883}\pm_{\mathbf{0.005}}$ | $0.725\pm_{0.043}$ | $\mathbf{0.848}\pm_{\mathbf{0.010}}$ | $0.632\pm_{0.047}$ | $\mathbf{0.881}\pm_{\mathbf{0.006}}$ |
| student-2M | $\underline{\mathbf{0.933}}\pm_{\mathbf{0.006}}$ | $\underline{\mathbf{0.882}}\pm_{\mathbf{0.007}}$ | $\underline{\mathbf{0.697}}\pm_{\mathbf{0.093}}$ | $\underline{\mathbf{0.893}}\pm_{\mathbf{0.002}}$ | $\underline{\mathbf{0.766}}\pm_{\mathbf{0.058}}$ | $\underline{\mathbf{0.868}}\pm_{\mathbf{0.010}}$ | $\underline{\mathbf{0.639}}\pm_{\mathbf{0.054}}$ | $\underline{\mathbf{0.892}}\pm_{\mathbf{0.004}}$ |

| | Tox AMES | Tox Carcinogens | Tox ClinTox | Tox DILI | Tox Skin R | Tox Tox21 | Tox hERG | Tox hERG (k) |
|---|---|---|---|---|---|---|---|---|
| 1-BertMTR | $0.862\pm_{0.014}$ | $\mathbf{0.831}\pm_{\mathbf{0.074}}$ | $0.601\pm_{0.069}$ | $0.831\pm_{0.060}$ | $\mathbf{0.815}\pm_{\mathbf{0.057}}$ | $0.801\pm_{0.056}$ | $0.826\pm_{0.023}$ | $0.877\pm_{0.012}$ |
| 1-3dinfo | $0.864\pm_{0.011}$ | $0.826\pm_{0.083}$ | $0.603\pm_{0.074}$ | $0.841\pm_{0.049}$ | $\underline{\mathbf{0.835}}\pm_{\mathbf{0.051}}$ | $0.802\pm_{0.055}$ | $\mathbf{0.831}\pm_{\mathbf{0.019}}$ | $0.875\pm_{0.007}$ |
| 2-Teachers | $\mathbf{0.875}\pm_{\mathbf{0.007}}$ | $0.827\pm_{0.062}$ | $\mathbf{0.657}\pm_{\mathbf{0.102}}$ | $\underline{\mathbf{0.865}}\pm_{\mathbf{0.024}}$ | $0.770\pm_{0.053}$ | $\mathbf{0.814}\pm_{\mathbf{0.057}}$ | $0.826\pm_{0.029}$ | $\mathbf{0.898}\pm_{\mathbf{0.006}}$ |
| student-2M | $\underline{\mathbf{0.891}}\pm_{\mathbf{0.014}}$ | $\underline{\mathbf{0.839}}\pm_{\mathbf{0.095}}$ | $\underline{\mathbf{0.662}}\pm_{\mathbf{0.072}}$ | $0.856\pm_{0.045}$ | $0.801\pm_{0.026}$ | $\underline{\mathbf{0.819}}\pm_{\mathbf{0.054}}$ | $\underline{\mathbf{0.834}}\pm_{\mathbf{0.019}}$ | $\underline{\mathbf{0.911}}\pm_{\mathbf{0.005}}$ |

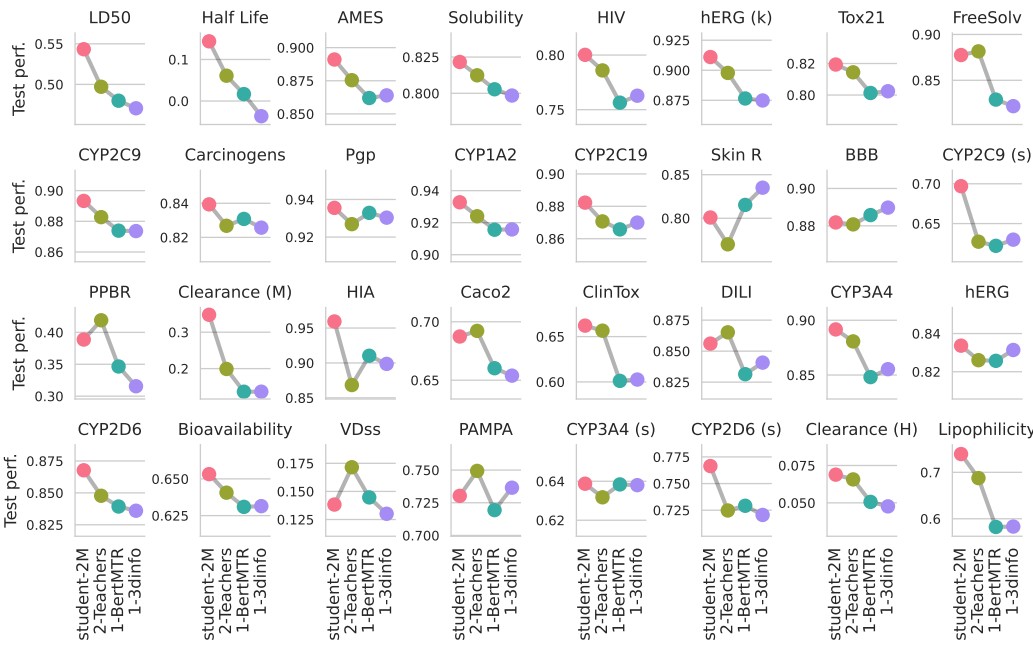

Figure 9: Test AUROC/$R^2$ score of the students on the classification/regression tasks, trained with all teachers (student-2M), two teachers (2-Teachers) and one teacher (1-ChemBertMTR for the model trained with ChemBertMTR-77M and 1-teacher-3dinfomax for the model trained with 3D-infomax).

## B.5 Comprehensive results

The following tables provide the raw results of the different evaluated models on the ADMET and HTS downstream tasks. Tab. 7 and Tab. 8 display the test performances of the models on the classification and regression tasks respectively. All regression tasks are evaluated using the $R^2$ score, while the classification tasks are evaluated using the AUROC score. We report the mean values of the metrics over 5 runs for each task, as well as the standard deviation.

We display in Figure 10 the evolution of the average rank of the embedders when separating the tasks based on the amount of samples, and the class imbalance (for classification tasks). Our student appears robust in both setups, even though as the class imbalance becomes more important, or as the amount of samples in the task decreases, the difference between the top-performing embedders becomes less significant.

Table 7: AUROC of each model on the ADMET and HTS downstream classification tasks. The best embedder for each task is highlighted in bold and underlined, and the second best is highlighted in bold.

| Model | Metabolism CYP3A4 | Metabolism CYP3A4 (s) | Metabolism CYP2D6 | Metabolism CYP2D6 (s) | Metabolism CYP2C9 | Metabolism CYP2C9 (s) | Metabolism CYP2C19 | Metabolism CYP1A2 | HTS HIV | Distribution BBB | avg |
|---|---|---|---|---|---|---|---|---|---|---|---|
| InfoGraph | 0.817 ± 0.003 | 0.567 ± 0.042 | 0.813 ± 0.013 | 0.674 ± 0.031 | 0.840 ± 0.007 | 0.624 ± 0.085 | 0.832 ± 0.004 | 0.878 ± 0.003 | 0.769 ± 0.018 | 0.843 ± 0.022 | 0.768 ± 0.097 |
| ChemGPT-1.2B | 0.844 ± 0.006 | 0.608 ± 0.026 | 0.816 ± 0.020 | 0.670 ± 0.014 | 0.852 ± 0.008 | 0.638 ± 0.034 | 0.835 ± 0.004 | 0.886 ± 0.008 | 0.760 ± 0.014 | 0.853 ± 0.012 | 0.779 ± 0.094 |
| FRAD QM9 (t) | 0.855 ± 0.003 | 0.607 ± 0.060 | 0.815 ± 0.018 | 0.711 ± 0.025 | 0.860 ± 0.006 | 0.609 ± 0.049 | 0.845 ± 0.010 | 0.906 ± 0.004 | 0.779 ± 0.005 | 0.869 ± 0.013 | 0.785 ± 0.111 |
| ChemBertMLM-10M | 0.846 ± 0.005 | 0.606 ± 0.029 | 0.813 ± 0.014 | 0.733 ± 0.017 | 0.851 ± 0.009 | 0.643 ± 0.020 | 0.843 ± 0.007 | 0.886 ± 0.007 | 0.733 ± 0.012 | 0.868 ± 0.009 | 0.787 ± 0.089 |
| GROVER | 0.827 ± 0.007 | 0.599 ± 0.028 | 0.824 ± 0.013 | 0.750 ± 0.040 | 0.853 ± 0.005 | 0.614 ± 0.042 | 0.843 ± 0.008 | 0.880 ± 0.006 | 0.760 ± 0.014 | 0.869 ± 0.016 | 0.787 ± 0.096 |
| GraphCL (t) | 0.846 ± 0.008 | 0.607 ± 0.044 | 0.828 ± 0.009 | 0.723 ± 0.030 | 0.862 ± 0.006 | 0.645 ± 0.032 | 0.858 ± 0.007 | 0.898 ± 0.004 | 0.765 ± 0.019 | 0.865 ± 0.011 | 0.792 ± 0.093 |
| GraphLog (t) | 0.847 ± 0.009 | 0.624 ± 0.049 | 0.832 ± 0.017 | 0.768 ± 0.046 | 0.860 ± 0.009 | 0.616 ± 0.034 | 0.855 ± 0.006 | 0.886 ± 0.006 | 0.748 ± 0.008 | 0.867 ± 0.017 | 0.790 ± 0.096 |
| GraphMVP (t) | 0.847 ± 0.011 | 0.619 ± 0.034 | 0.830 ± 0.016 | 0.747 ± 0.033 | 0.870 ± 0.005 | 0.622 ± 0.082 | 0.865 ± 0.006 | 0.899 ± 0.004 | 0.771 ± 0.016 | 0.874 ± 0.016 | 0.800 ± 0.097 |
| MolR gat | 0.873 ± 0.005 | 0.598 ± 0.045 | 0.841 ± 0.017 | 0.717 ± 0.017 | 0.869 ± 0.004 | 0.615 ± 0.037 | 0.859 ± 0.010 | 0.909 ± 0.004 | 0.797 ± 0.012 | 0.867 ± 0.025 | 0.808 ± 0.098 |
| ThreeDInfomax (t) | 0.858 ± 0.009 | 0.604 ± 0.015 | 0.842 ± 0.010 | 0.747 ± 0.039 | 0.865 ± 0.009 | 0.571 ± 0.085 | 0.868 ± 0.008 | 0.917 ± 0.005 | 0.762 ± 0.010 | 0.885 ± 0.019 | 0.815 ± 0.100 |
| ChemBertMTR-77M (t) | 0.883 ± 0.008 | 0.645 ± 0.051 | 0.845 ± 0.015 | 0.734 ± 0.024 | 0.877 ± 0.007 | 0.600 ± 0.114 | 0.873 ± 0.005 | 0.919 ± 0.007 | 0.797 ± 0.014 | 0.894 ± 0.026 | 0.816 ± 0.096 |
| MSE | 0.871 ± 0.008 | 0.641 ± 0.063 | 0.849 ± 0.012 | 0.742 ± 0.034 | 0.873 ± 0.006 | 0.663 ± 0.049 | 0.864 ± 0.008 | 0.917 ± 0.007 | 0.797 ± 0.005 | 0.878 ± 0.022 | 0.806 ± 0.094 |
| Cosine | 0.892 ± 0.005 | 0.637 ± 0.025 | 0.855 ± 0.015 | 0.758 ± 0.045 | 0.887 ± 0.008 | 0.673 ± 0.070 | 0.879 ± 0.005 | 0.925 ± 0.003 | 0.786 ± 0.014 | 0.881 ± 0.014 | 0.816 ± 0.096 |
| student-250k | 0.893 ± 0.009 | 0.646 ± 0.052 | 0.865 ± 0.017 | 0.767 ± 0.040 | 0.893 ± 0.005 | 0.699 ± 0.080 | 0.881 ± 0.006 | 0.931 ± 0.005 | 0.796 ± 0.013 | 0.875 ± 0.020 | 0.823 ± 0.095 |
| student-2M | 0.892 ± 0.004 | 0.639 ± 0.054 | 0.868 ± 0.010 | 0.766 ± 0.058 | 0.893 ± 0.002 | 0.697 ± 0.093 | 0.882 ± 0.007 | 0.933 ± 0.006 | 0.800 ± 0.014 | 0.882 ± 0.020 | 0.825 ± 0.096 |

| Model | Tox hERG (k) | Tox hERG | Tox Tox21 | Tox Skin R | Tox DILI | Tox ClinTox | Tox Carcinogens | Tox AMES | Absorption Pgp | Absorption PAMPA | Absorption HIA | Absorption Bioavailability |
|---|---|---|---|---|---|---|---|---|---|---|---|---|
| InfoGraph | 0.849 ± 0.009 | 0.778 ± 0.027 | 0.770 ± 0.058 | 0.714 ± 0.030 | 0.837 ± 0.056 | 0.621 ± 0.086 | 0.728 ± 0.042 | 0.853 ± 0.009 | 0.896 ± 0.022 | 0.685 ± 0.031 | 0.872 ± 0.085 | 0.631 ± 0.015 |
| ChemGPT-1.2B | 0.867 ± 0.007 | 0.789 ± 0.049 | 0.762 ± 0.066 | 0.721 ± 0.077 | 0.857 ± 0.031 | 0.641 ± 0.022 | 0.785 ± 0.017 | 0.843 ± 0.012 | 0.926 ± 0.027 | 0.665 ± 0.050 | 0.859 ± 0.055 | 0.668 ± 0.046 |
| FRAD QM9 (t) | 0.873 ± 0.004 | 0.817 ± 0.032 | 0.797 ± 0.060 | 0.747 ± 0.057 | 0.843 ± 0.044 | 0.553 ± 0.054 | 0.772 ± 0.057 | 0.871 ± 0.009 | 0.914 ± 0.024 | 0.699 ± 0.043 | 0.945 ± 0.034 | 0.626 ± 0.022 |
| ChemBertMLM-10M | 0.867 ± 0.007 | 0.779 ± 0.017 | 0.789 ± 0.065 | 0.747 ± 0.065 | 0.791 ± 0.081 | 0.648 ± 0.082 | 0.776 ± 0.063 | 0.858 ± 0.013 | 0.911 ± 0.029 | 0.715 ± 0.056 | 0.892 ± 0.082 | 0.664 ± 0.069 |
| GROVER | 0.856 ± 0.004 | 0.774 ± 0.034 | 0.780 ± 0.059 | 0.749 ± 0.081 | 0.844 ± 0.036 | 0.637 ± 0.053 | 0.779 ± 0.084 | 0.867 ± 0.012 | 0.918 ± 0.029 | 0.703 ± 0.027 | 0.931 ± 0.038 | 0.643 ± 0.027 |
| GraphCL (t) | 0.864 ± 0.005 | 0.799 ± 0.038 | 0.787 ± 0.058 | 0.770 ± 0.038 | 0.827 ± 0.085 | 0.639 ± 0.078 | 0.847 ± 0.064 | 0.869 ± 0.016 | 0.920 ± 0.030 | 0.709 ± 0.034 | 0.863 ± 0.052 | 0.622 ± 0.071 |
| GraphLog (t) | 0.849 ± 0.006 | 0.797 ± 0.049 | 0.801 ± 0.052 | 0.751 ± 0.10 | 0.853 ± 0.035 | 0.696 ± 0.081 | 0.793 ± 0.076 | 0.869 ± 0.006 | 0.920 ± 0.026 | 0.637 ± 0.024 | 0.897 ± 0.035 | 0.694 ± 0.055 |
| GraphMVP (t) | 0.872 ± 0.005 | 0.823 ± 0.045 | 0.793 ± 0.085 | 0.750 ± 0.087 | 0.867 ± 0.049 | 0.624 ± 0.046 | 0.779 ± 0.095 | 0.874 ± 0.011 | 0.918 ± 0.038 | 0.718 ± 0.009 | 0.944 ± 0.055 | 0.672 ± 0.049 |
| MolR gat | 0.881 ± 0.011 | 0.844 ± 0.022 | 0.800 ± 0.061 | 0.748 ± 0.047 | 0.858 ± 0.042 | 0.810 ± 0.048 | 0.760 ± 0.057 | 0.871 ± 0.014 | 0.928 ± 0.028 | 0.705 ± 0.061 | 0.957 ± 0.020 | 0.670 ± 0.033 |
| ThreeDInfomax (t) | 0.874 ± 0.010 | 0.829 ± 0.035 | 0.804 ± 0.059 | 0.833 ± 0.041 | 0.842 ± 0.037 | 0.837 ± 0.043 | 0.791 ± 0.074 | 0.872 ± 0.018 | 0.929 ± 0.030 | 0.745 ± 0.026 | 0.986 ± 0.014 | 0.683 ± 0.027 |
| ChemBertMTR-77M (t) | 0.897 ± 0.009 | 0.832 ± 0.034 | 0.818 ± 0.063 | 0.758 ± 0.069 | 0.858 ± 0.037 | 0.734 ± 0.068 | 0.776 ± 0.033 | 0.881 ± 0.005 | 0.936 ± 0.030 | 0.763 ± 0.026 | 0.960 ± 0.034 | 0.626 ± 0.076 |
| MSE | 0.895 ± 0.007 | 0.824 ± 0.018 | 0.807 ± 0.061 | 0.770 ± 0.039 | 0.856 ± 0.033 | 0.654 ± 0.094 | 0.783 ± 0.039 | 0.871 ± 0.010 | 0.914 ± 0.030 | 0.735 ± 0.027 | 0.914 ± 0.040 | 0.629 ± 0.043 |
| Cosine | 0.908 ± 0.006 | 0.830 ± 0.038 | 0.814 ± 0.056 | 0.780 ± 0.033 | 0.879 ± 0.030 | 0.650 ± 0.092 | 0.822 ± 0.084 | 0.884 ± 0.008 | 0.926 ± 0.021 | 0.755 ± 0.024 | 0.908 ± 0.062 | 0.671 ± 0.043 |
| student-250k | 0.912 ± 0.006 | 0.828 ± 0.050 | 0.817 ± 0.062 | 0.812 ± 0.040 | 0.858 ± 0.036 | 0.633 ± 0.082 | 0.810 ± 0.079 | 0.888 ± 0.009 | 0.926 ± 0.025 | 0.742 ± 0.025 | 0.969 ± 0.018 | 0.653 ± 0.055 |
| student-2M | 0.911 ± 0.005 | 0.834 ± 0.019 | 0.819 ± 0.054 | 0.801 ± 0.026 | 0.856 ± 0.045 | 0.662 ± 0.072 | 0.839 ± 0.095 | 0.891 ± 0.014 | 0.936 ± 0.024 | 0.730 ± 0.024 | 0.959 ± 0.026 | |

Table 8: $R^2$ score of each model on the ADMET downstream regression tasks. The best embedder for each task is highlighted in bold and underlined, and the second best is highlighted in bold.

| | avg | Absorption Caco2 | Absorption FreeSolv | Absorption Lipophilicity | Absorption Solubility |
|---|---|---|---|---|---|
| InfoGraph | $0.275 \pm 0.284$ | $0.491 \pm 0.031$ | $0.639 \pm 0.058$ | $0.341 \pm 0.035$ | $0.700 \pm 0.007$ |
| ChemBertMLM-10M | $0.264 \pm 0.364$ | $0.543 \pm 0.076$ | $0.776 \pm 0.038$ | $0.363 \pm 0.063$ | $0.774 \pm 0.007$ |
| FRAD QM9[t] | $0.332 \pm 0.284$ | $0.564 \pm 0.051$ | $0.686 \pm 0.082$ | $0.483 \pm 0.029$ | $0.758 \pm 0.011$ |
| ChemGPT-1.2B | $0.340 \pm 0.329$ | $0.567 \pm 0.079$ | $0.831 \pm 0.048$ | $0.487 \pm 0.020$ | $0.798 \pm 0.009$ |
| GROVER | $0.350 \pm 0.274$ | $0.575 \pm 0.058$ | $0.708 \pm 0.024$ | $0.470 \pm 0.043$ | $0.733 \pm 0.027$ |
| GraphLog[t] | $0.350 \pm 0.311$ | $0.545 \pm 0.055$ | $0.811 \pm 0.017$ | $0.486 \pm 0.037$ | $0.765 \pm 0.010$ |
| GraphCL[t] | $0.355 \pm 0.292$ | $0.559 \pm 0.051$ | $0.764 \pm 0.038$ | $0.467 \pm 0.067$ | $0.745 \pm 0.021$ |
| GraphMVP[t] | $0.397 \pm 0.320$ | $0.592 \pm 0.064$ | $0.861 \pm 0.036$ | $0.590 \pm 0.064$ | $0.791 \pm 0.009$ |
| MolR gat | $0.394 \pm 0.307$ | $0.651 \pm 0.089$ | $0.804 \pm 0.075$ | $0.518 \pm 0.037$ | $0.822 \pm 0.010$ |
| ThreeDInfomax[t] | $0.425 \pm 0.322$ | $0.700 \pm 0.038$ | $0.852 \pm 0.055$ | $0.624 \pm 0.031$ | **$\underline{0.848} \pm \underline{0.004}$** |
| ChemBertMTR-77M[t] | $0.459 \pm 0.308$ | **$\underline{0.725} \pm \underline{0.027}$** | $0.874 \pm 0.037$ | $0.670 \pm 0.025$ | **$0.839 \pm 0.007$** |
| MSE | $0.420 \pm 0.299$ | $0.642 \pm 0.060$ | $0.851 \pm 0.063$ | $0.605 \pm 0.021$ | $0.792 \pm 0.018$ |
| Cosine | $0.460 \pm 0.311$ | $0.699 \pm 0.056$ | **$0.893 \pm 0.034$** | $0.721 \pm 0.028$ | $0.815 \pm 0.009$ |
| student-250k | **$\underline{0.482} \pm \underline{0.298}$** | **$0.712 \pm 0.040$** | **$\underline{0.900} \pm \underline{0.035}$** | **$\underline{0.742} \pm \underline{0.019}$** | $0.823 \pm 0.007$ |
| student-2M | **$0.476 \pm 0.301$** | $0.687 \pm 0.045$ | $0.878 \pm 0.036$ | **$0.739 \pm 0.021$** | $0.822 \pm 0.005$ |

| | Distribution PPBR | Distribution VDss | Excretion Clearance (H) | Excretion Clearance (M) | Excretion Half Life | Tox LD50 |
|---|---|---|---|---|---|---|
| InfoGraph | $0.093 \pm 0.073$ | $0.018 \pm 0.190$ | $-0.048 \pm 0.133$ | $0.070 \pm 0.046$ | $-0.011 \pm 0.161$ | $0.458 \pm 0.039$ |
| ChemBertMLM-10M | $0.112 \pm 0.035$ | $0.066 \pm 0.091$ | $-0.185 \pm 0.122$ | $0.040 \pm 0.178$ | $-0.240 \pm 0.279$ | $0.390 \pm 0.044$ |
| FRAD QM9[t] | $0.180 \pm 0.031$ | $-0.004 \pm 0.050$ | $0.006 \pm 0.095$ | $0.124 \pm 0.059$ | $0.104 \pm 0.129$ | $0.415 \pm 0.039$ |
| ChemGPT-1.2B | $0.175 \pm 0.036$ | $0.046 \pm 0.173$ | $-0.018 \pm 0.071$ | $0.117 \pm 0.099$ | $-0.047 \pm 0.182$ | $0.442 \pm 0.043$ |
| GROVER | $0.185 \pm 0.056$ | **$0.186 \pm 0.079$** | $-0.034 \pm 0.095$ | $0.197 \pm 0.082$ | $0.035 \pm 0.161$ | $0.447 \pm 0.058$ |
| GraphLog[t] | $0.240 \pm 0.082$ | **$\underline{0.202} \pm \underline{0.111}$** | $-0.094 \pm 0.053$ | $0.068 \pm 0.120$ | $0.018 \pm 0.192$ | $0.457 \pm 0.054$ |
| GraphCL[t] | $0.237 \pm 0.048$ | $0.158 \pm 0.075$ | $-0.022 \pm 0.127$ | $0.123 \pm 0.108$ | $0.007 \pm 0.165$ | $0.508 \pm 0.026$ |
| GraphMVP[t] | $0.327 \pm 0.036$ | $0.168 \pm 0.081$ | $-0.009 \pm 0.135$ | $0.144 \pm 0.071$ | $-0.017 \pm 0.226$ | $0.527 \pm 0.042$ |
| MolR gat | $0.284 \pm 0.093$ | $0.155 \pm 0.180$ | $-0.024 \pm 0.091$ | $0.174 \pm 0.050$ | $0.059 \pm 0.232$ | $0.496 \pm 0.040$ |
| ThreeDInfomax[t] | $0.314 \pm 0.053$ | $0.152 \pm 0.061$ | $0.071 \pm 0.049$ | $0.195 \pm 0.114$ | $-0.004 \pm 0.264$ | $0.500 \pm 0.040$ |
| ChemBertMTR-77M[t] | **$0.393 \pm 0.055$** | $0.138 \pm 0.127$ | $0.011 \pm 0.048$ | $0.250 \pm 0.078$ | **$0.196 \pm 0.190$** | $0.491 \pm 0.031$ |
| MSE | $0.362 \pm 0.077$ | $0.135 \pm 0.097$ | $0.034 \pm 0.097$ | $0.244 \pm 0.062$ | $0.060 \pm 0.116$ | $0.470 \pm 0.030$ |
| Cosine | $0.382 \pm 0.032$ | $0.108 \pm 0.084$ | **$0.079 \pm 0.102$** | $0.275 \pm 0.054$ | $0.111 \pm 0.158$ | $0.515 \pm 0.039$ |
| student-250k | **$0.390 \pm 0.042$** | $0.125 \pm 0.111$ | **$\underline{0.113} \pm \underline{0.070}$** | $0.283 \pm 0.076$ | **$\underline{0.207} \pm \underline{0.101}$** | **$0.529 \pm 0.039$** |
| student-2M | $0.389 \pm 0.050$ | $0.138 \pm 0.115$ | $0.069 \pm 0.060$ | **$\underline{0.348} \pm \underline{0.062}$** | $0.144 \pm 0.205$ | **$\underline{0.543} \pm \underline{0.041}$** |

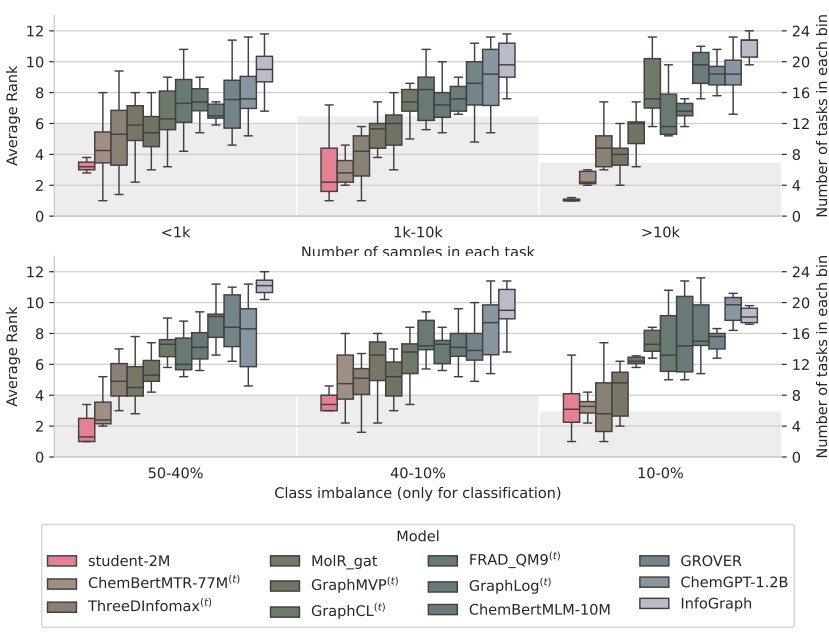

Figure 10: Average ranking of our models when grouping tasks based on the number of samples in the task and the class imbalance (for classification tasks).

# C Natural Language Processing

## C.1 Training set and hyperparameters

### C.1.1 Training set

**Dataset sources.** We ran experiments with two training sets a home-made dataset combining different training sets of different embedders and the GISTEmbed dataset. We provide the statistics of our dataset in Tab. 9 and the GISTEmbed dataset is described in (Solatorio, 2024).

**Dataset construction.** Most embedding datasets consists of positive and negative samples, questions and answers, or sentences and their labels. We flattened the datasets to have only one column of sentences and deduplicated the dataset. For the MEDI () dataset for example, given query, positive and negative samples we build a dataset with three times the number of entries, one for each sentence. We then deduplicated the dataset to remove any duplicate entries.

Table 9: Number of samples in each dataset

| URL | Number of samples |
|---|---|
| https://huggingface.co/datasets/embedding-data/SPECTER | 190872 |
| https://huggingface.co/datasets/embedding-data/Amazon-QA | 3264474 |
| https://huggingface.co/datasets/embedding-data/simple-wiki | 203755 |
| https://huggingface.co/datasets/embedding-data/QQP_triplets | 328188 |
| https://huggingface.co/datasets/embedding-data/sentence-compression | 356409 |
| https://huggingface.co/datasets/embedding-data/altlex | 223901 |
| https://huggingface.co/datasets/fancyzhx/ag_news | 120000 |
| https://huggingface.co/datasets/stanfordnlp/sst2 | 67349 |
| https://huggingface.co/datasets/dair-ai/emotion | 416809 |
| https://huggingface.co/datasets/stanfordnlp/snli | 1100304 |
| https://huggingface.co/datasets/cardiffnlp/tweet_eval | 45000 |
| https://huggingface.co/datasets/stanfordnlp/imdb | 25000 |
| | 6342061 |

Table 10: Performance of the 4 teachers we used and of the base students. Experiments with single teacher distillation were performed with the stronger teacher SFR-Embedding-2_R.

| | | Size | Amazon Counterfactual | Amazon Polarity | Amazon Reviews | Banking77 | Emotion | Imdb | MTOPDomain | MTOPIntent | Massive Intent | Massive Scenario | Toxic Conversations | Tweet Sentiment Extraction | Avg. |
|---|---|---|---|---|---|---|---|---|---|---|---|---|---|---|---|
| Teacher | SFR-Embedding-2_R | 7111.0 | 92.7 | 97.3 | 61.0 | 90.0 | 93.4 | 96.8 | 98.6 | 91.3 | 86.0 | 90.6 | 91.1 | 79.7 | 89.0 |
| | stella_en_400M_v5 | 435.0 | 92.4 | 97.2 | 59.5 | 89.3 | 78.8 | 96.5 | 98.8 | 92.3 | 85.2 | 89.6 | 86.9 | 73.6 | 86.7 |
| | UAE-Large-V1 | 335.0 | 75.5 | 92.8 | 48.3 | 87.7 | 51.8 | 92.8 | 94.0 | 76.9 | 76.5 | 79.8 | 71.1 | 59.8 | 75.6 |
| | sf_model_e5 | 335.0 | 70.8 | 91.8 | 48.9 | 84.6 | 54.9 | 93.1 | 93.6 | 66.0 | 73.5 | 77.4 | 71.2 | 61.5 | 74.0 |
| Student (Base) | snowflake-arctic-embed-m | 109.0 | 76.8 | 82.8 | 38.9 | 80.3 | 46.5 | 74.1 | 92.7 | 65.2 | 66.9 | 72.8 | 64.9 | 56.7 | 68.2 |
| | snowflake-arctic-embed-s | 33.0 | 71.2 | 78.8 | 38.3 | 79.1 | 45.8 | 69.5 | 90.9 | 58.6 | 64.8 | 70.0 | 62.0 | 58.9 | 65.7 |
| | snowflake-arctic-embed-xs | 23.0 | 65.1 | 70.0 | 35.3 | 76.4 | 41.8 | 62.8 | 90.8 | 58.0 | 63.5 | 71.0 | 64.3 | 56.2 | 62.9 |

### C.1.2 Teachers and based students performance

**Teachers.** We selected 4 teachers from the MTEB benchmark (Muennighoff et al., 2023) as teachers for our distillation method. We provide the list of the teachers and their performance in Tab. 10. The 4 teachers of widely different sizes (335M, 435M and 7B) have display strong but different performances on the MTEB benchmark.

### C.1.3 Single teacher distillation

**Single teacher vs. Multi-Teachers.** Since some teachers yield strong performance on their own, distilling only from the strongest could yield similar results as the multi-teacher setting involving weaker teachers. We applied our method in a single-teacher setting using the strongest teacher by far (SF-Embeddings-R_2) as a teacher and compared the results to the multi-teacher setting. Consistently with results in computer vision and molecular representations, we found that adding weaker teachers did improve our results (Figure 11), supporting our hypothesis that enforcing reconstruction capabilities for a diversity of models indeed leads to more informative representations.

### C.1.4 Hyperparameters

**Training hyperparameters.** We trained our models using the Adam optimizer with a constant learning rate of $5.10^{-5}$ and an effective batch size of 16 for all our models.

Figure 11: Comparison of distilled small model with the performance of the initial backbone, baselines in the MTEB, with our teachers' performance.

### C.2 Detailed evaluation results

We ran different parts of the MTEB benchmarks and report the overall results for all our models in this section.

### C.2.1 Evaluation on classification tasks

**Small models' performance.** In Tab. 11 and Tab. 12, we provide the classification accuracy of our distilled models on the MTEB classification benchmark for our smaller models xs (22M) and s (33M). Our smallest model significantly improves SOTA performance for models of its size by increasing the average score of 2 points compared to the previous best model.

### C.2.2 Evaluation on similarity and clustering tasks

**Limited structure of our embedding spaces.** Our method only seeks to pack as much (statistical) information into the embeddings as possible without any constraints on the underlying structure of the embedding space. It is therefore not surprising that methods that relies on metrics on the embedding space such as similarity tasks do not perform as well as the classification tasks. However,

Table 11: Performance of our distilled models compared to models of similar sizes 16M to 30M parameters from the MTEB Benchmark on classification tasks.

| Task | | Size | Amazon Counterfactual | Amazon Polarity | Amazon Reviews | Banking77 | Emotion | Imdb | MTOPDomain | MTOPIntent | Massive Intent | Massive Scenario | Toxic Conversations | Tweet Sentiment Extraction | Avg. |
|---|---|---|---|---|---|---|---|---|---|---|---|---|---|---|---|
| Model | | | | | | | | | | | | | | | |
| MTEB | GIST | 23M | 72.9 | **87.2** | 42.6 | 84.2 | 52.1 | 78.5 | 94.8 | 77.7 | 73.2 | 76.7 | 72.9 | 59.9 | 72.7 |
| | Bulbasaur | 17M | 71.9 | 78.8 | 39.3 | 80.6 | 44.8 | 71.5 | 90.8 | 68.7 | 68.8 | 73.8 | 66.3 | 59.5 | 67.9 |
| | Ivysaur | 23M | 72.1 | 86.7 | 42.7 | 81.9 | 45.4 | 80.8 | 92.1 | 71.9 | 70.3 | 74.9 | 65.5 | 58.7 | 70.2 |
| | Squirtle | 16M | 69.6 | 82.1 | 41.9 | 67.1 | 45.8 | 75.0 | 87.3 | 54.7 | 61.5 | 67.0 | 64.5 | 61.8 | 64.9 |
| | Venusaur | 16M | 73.2 | 80.0 | 39.7 | 78.0 | 44.4 | 73.0 | 89.9 | 71.0 | 67.8 | 72.4 | 64.4 | 59.7 | 67.8 |
| | Wartortle | 17M | 70.4 | 82.0 | 42.4 | 71.1 | 46.8 | 74.6 | 88.2 | 54.9 | 62.3 | 68.2 | 65.2 | **62.5** | 65.7 |
| | gte-micro | 17M | 68.8 | 77.1 | 40.9 | 69.6 | 46.2 | 62.2 | 86.7 | 49.7 | 59.0 | 66.6 | 66.1 | 60.8 | 62.8 |
| | gte-micro-v2 | 17M | 71.4 | 77.7 | 39.0 | 80.4 | 44.5 | 70.6 | 90.5 | 67.5 | 68.5 | 73.5 | 66.7 | 59.3 | 67.5 |
| | gte-micro-v4 | 19M | 71.8 | 80.0 | 39.8 | 80.9 | 44.9 | 72.0 | 90.9 | 68.5 | 69.1 | 74.2 | 66.0 | 59.4 | 68.1 |
| | snowflake-arctic-embed-xs | 23M | 65.1 | 70.0 | 35.3 | 76.4 | 41.8 | 62.8 | 90.8 | 58.0 | 63.5 | 71.0 | 64.3 | 56.2 | 62.9 |
| | bge-micro | 17M | 66.3 | 75.4 | 35.8 | 80.6 | 42.5 | 70.7 | 90.2 | 68.0 | 67.8 | 73.0 | 69.2 | 56.7 | 66.3 |
| | bge-micro-v2 | 17M | 67.8 | 79.8 | 37.5 | 81.2 | 44.5 | 76.5 | 90.7 | 68.3 | 68.6 | 73.9 | 70.2 | 57.6 | 68.0 |
| | gte-tiny | 23M | 71.8 | 86.6 | 42.6 | 81.7 | 44.7 | 80.5 | 91.8 | 69.9 | 70.1 | 74.9 | 71.0 | 58.6 | 70.3 |
| | slx-v0.1 | 23M | 61.5 | 64.3 | 30.3 | 80.0 | 40.5 | 61.8 | 92.0 | 63.3 | 73.9 | 72.1 | 54.0 | | 62.6 |
| | multi-qa-MiniLM-L6-cos-v1 | 23M | 61.8 | 62.4 | 29.6 | 78.6 | 39.6 | 61.2 | 90.0 | 59.6 | 66.8 | 73.8 | 65.1 | 51.6 | 61.7 |
| | all-MiniLM-L6-v2 | 23M | 63.6 | 64.3 | 30.9 | 80.0 | 40.8 | 61.8 | 91.7 | 61.5 | 66.9 | 73.8 | 62.1 | 54.0 | 62.6 |
| MSE | Student-xs | 23M | 71.6 | 86.2 | 42.3 | 83.6 | 57.5 | **83.5** | 94.5 | 75.4 | 74.3 | **80.4** | 66.3 | 59.3 | 72.9 |
| NLL | Student-xs | 23M | **76.5** | 84.9 | 42.4 | **85.8** | **58.0** | 81.1 | **95.2** | **79.9** | **75.8** | 80.4 | 68.1 | 60.1 | **74.0** |

Table 12: Performance of our distilled models compared to models of similar sizes 30M to 50M parameters from the MTEB Benchmark on classification tasks.

| Task | | Size | Amazon Counterfactual | Amazon Polarity | Amazon Reviews | Banking77 | Emotion | Imdb | MTOPDomain | MTOPIntent | Massive Intent | Massive Scenario | Toxic Conversations | Tweet Sentiment Extraction | Avg. |
|---|---|---|---|---|---|---|---|---|---|---|---|---|---|---|---|
| Model | | | | | | | | | | | | | | | |
| MTEB | bge-small-en-v1.5 | 33M | 73.8 | 92.8 | 47.0 | 85.7 | 47.8 | **90.6** | 93.4 | 74.8 | 74.8 | 78.7 | 69.9 | 60.5 | 74.1 |
| | GIST | 33M | 75.3 | 93.2 | 49.7 | 86.7 | 55.9 | 89.5 | 95.5 | 79.1 | 75.5 | 79.2 | 72.8 | 61.0 | **76.1** |
| | NoInstruct | 33M | 75.8 | **93.3** | **50.0** | 86.4 | 55.1 | 90.2 | 95.3 | 79.6 | 79.3 | | 69.4 | 61.3 | 76.0 |
| | snowflake-arctic-embed-s | 33M | 71.2 | 78.8 | 38.3 | 79.1 | 45.8 | 69.5 | 90.9 | 58.6 | 64.8 | 70.0 | 62.0 | 58.9 | 65.7 |
| | bge-small-4096 | 35M | 68.8 | 81.3 | 38.6 | 80.0 | 40.1 | 80.1 | 90.4 | 66.5 | 67.6 | 73.5 | 69.3 | 57.6 | 67.8 |
| | LASER | 43M | 76.8 | 61.0 | 28.7 | 67.8 | 24.8 | 57.6 | 75.4 | 49.5 | 47.9 | 55.9 | 54.0 | 48.7 | 53.2 |
| | e5-small | 33M | 76.2 | 87.5 | 42.6 | 81.9 | 46.9 | 75.5 | 92.0 | 73.2 | 72.2 | 75.8 | 72.8 | **63.3** | 71.7 |
| | e5-small-v2 | 33M | **77.6** | 91.3 | 45.9 | 81.6 | 47.1 | 86.0 | 92.7 | 72.6 | 71.6 | 76.4 | 71.1 | 61.5 | 72.9 |
| | jina-embedding-s-en-v1 | 35M | 64.8 | 64.3 | 30.6 | 74.6 | 36.1 | 58.7 | 88.8 | 58.6 | 64.7 | 71.8 | 59.4 | 54.3 | 60.6 |
| | jina-embeddings-v2-small-en | 33M | 71.4 | 82.9 | 40.9 | 78.2 | 44.0 | 73.6 | 94.0 | 72.5 | 69.8 | 71.5 | 59.4 | 54.3 | 68.8 |
| | all-MiniLM-L12-v2 | 33M | 65.3 | 63.0 | 30.8 | 80.4 | 41.2 | 59.8 | 91.9 | 62.8 | 67.2 | 74.6 | 67.5 | 54.2 | 63.2 |
| | gte-small | 33M | 73.2 | 91.8 | 48.0 | 84.1 | 46.6 | 86.8 | 93.0 | 69.7 | 70.3 | 75.6 | 70.3 | 58.2 | 72.3 |
| MSE | Student-s | 33M | 72.6 | 90.3 | 44.3 | 84.2 | 56.5 | 88.8 | 94.9 | 77.2 | 75.4 | **81.2** | 64.9 | 60.4 | 74.2 |
| NLL | Student-s | 33M | 77.3 | 89.2 | 43.8 | **86.7** | **58.0** | 88.3 | **95.5** | **81.9** | **76.7** | 80.7 | 66.1 | 60.6 | 75.4 |

our embedder are still competitive on these tasks achieving average performance for their respective size categories.

**Clustering with very small model.** In Tab. 15, we show that our very small model actually outperforms baselines and sits on the pareto frontier for clustering tasks. This is a surprising result as we did not optimize our models for clustering tasks and the embeddings are not designed to have a meaningful structure.

### C.2.3 Analysis and compare with the most recent embedders

The results at Tab. 22 show that our medium model (STUDENT-M-NLL, 109M) achieves an average of 80.2 on the selected MTEB classification tasks, tracking much larger recent embedders within single-digit margins. In particular, QWEN3-EMBEDDING-0.6B (595M) reaches 85.8, a +5.6 point gain at $\sim 5.5\times$ the parameters. Substantially larger improvements appear only beyond $\sim$1B parameters (JASPER_EN_VISION_LANGUAGE_V1, 1.0B: 90.3; STELLA_EN_1.5B_V5, 1.5B: 89.4; QWEN3-EMBEDDING-4B, 4.0B: 89.8). Overall, the 109M model delivers competitive accuracy relative to 4–6× larger embedders, supporting our claim that multi-teacher distillation yields high information density at compact scales.

Table 13: Performance of our distilled models compared to models of similar sizes 100M to 120M parameters from the MTEB Benchmark on classification tasks.

| Task | Model | Size | Amazon Counterfactual | Amazon Polarity | Amazon Reviews | Banking77 | Emotion | Imdb | MTOPDomain | MTOPIntent | Massive Intent | Massive Scenario | Toxic Conversations | Tweet Sentiment Extraction | Avg. |
|---|---|---|---|---|---|---|---|---|---|---|---|---|---|---|---|
| MTEB | bge-base-en-v1.5 | 109M | 76.2 | 93.4 | 48.9 | 87.0 | 51.9 | 90.8 | 94.2 | 76.9 | 76.2 | 80.2 | 71.6 | 59.4 | 75.5 |
| | GIST | 109M | 76.0 | 93.5 | 50.5 | 87.3 | 54.7 | 89.7 | 95.3 | 78.1 | 76.0 | 79.6 | 72.4 | 59.3 | 76.0 |
| | bilingual-embedding-small | 118M | 74.3 | 82.2 | 40.2 | 80.3 | 40.8 | 73.7 | 89.7 | 66.5 | 68.9 | 74.5 | 62.5 | 59.6 | 67.8 |
| | multilingual-e5-small | 118M | 73.8 | 88.7 | 44.7 | 79.4 | 42.5 | 80.8 | 91.1 | 71.1 | 70.3 | 74.5 | 69.4 | 62.6 | 70.7 |
| | snowflake-arctic-embed-m | 109M | 76.8 | 82.8 | 38.9 | 80.3 | 46.5 | 74.1 | 92.7 | 65.2 | 66.9 | 72.8 | 64.9 | 56.7 | 68.2 |
| | snowflake-arctic-embed-m-v1.5 | 109M | 68.3 | 90.3 | 46.3 | 80.0 | 43.7 | 84.4 | 91.4 | 60.6 | 66.7 | 73.1 | 66.8 | 53.9 | 68.8 |
| | ml-nlp-elser.html | 110M | 74.2 | 61.9 | 32.1 | 82.0 | 46.6 | 65.0 | 93.2 | 71.1 | 68.5 | 75.0 | 68.2 | 53.6 | 65.9 |
| | e5-base-4k | 112M | 77.8 | 92.8 | 46.7 | 83.5 | 47.0 | 86.2 | 93.7 | 75.3 | 73.0 | 77.7 | 72.1 | 60.4 | 73.8 |
| | instructor-base | 110M | **86.2** | 88.4 | 44.6 | 77.0 | 51.8 | 81.2 | 93.7 | 70.3 | 67.5 | 72.6 | 71.8 | **63.3** | 72.4 |
| | bert-base-uncased | 110M | 74.2 | 71.3 | 33.6 | 63.4 | 35.3 | 65.3 | 82.6 | 68.1 | 59.9 | 64.3 | 70.0 | 51.8 | 61.7 |
| | e5-base | 109M | 79.7 | 88.0 | 42.6 | 83.3 | 49.4 | 76.0 | 93.2 | 74.8 | 72.2 | 76.8 | **74.1** | 61.4 | 72.6 |
| | e5-base-v2 | 110M | 77.8 | 92.8 | 46.7 | 83.5 | 47.0 | 86.2 | 93.7 | 75.3 | 73.0 | 77.7 | 72.1 | 60.4 | 73.8 |
| | jina-embedding-b-en-v1 | 110M | 66.7 | 67.6 | 31.2 | 84.1 | 44.7 | 63.9 | 91.5 | 72.8 | 71.1 | 76.2 | 66.2 | 56.9 | 66.6 |
| | contriever-base-msmarco | 110M | 72.2 | 68.6 | 37.4 | 80.0 | 44.8 | 67.0 | 93.2 | 69.3 | 67.8 | 76.0 | 67.8 | 56.1 | 66.7 |
| | sup-simcse-bert-base-uncased | 110M | 75.8 | 82.5 | 39.6 | 75.8 | 44.8 | 73.5 | 84.3 | 63.1 | 66.0 | 70.8 | 72.0 | 59.7 | 67.3 |
| | unsup-simcse-bert-base-uncased | 110M | 67.1 | 74.5 | 33.9 | 73.5 | 42.2 | 69.6 | 81.7 | 59.2 | 59.8 | 66.2 | 68.8 | 53.4 | 62.5 |
| | all-mpnet-base-v2 | 110M | 65.0 | 67.1 | 31.4 | 81.7 | 42.2 | 71.2 | 91.9 | 68.3 | 69.8 | 75.7 | 61.0 | 55.0 | 65.0 |
| | allenai-specter | 110M | 58.7 | 57.8 | 26.3 | 66.7 | 24.8 | 56.4 | 74.5 | 50.0 | 51.7 | 58.6 | 57.4 | 45.5 | 52.4 |
| | gtr-t5-base | 110M | 69.3 | 67.8 | 38.5 | 79.3 | 42.2 | 66.0 | 92.4 | 62.4 | 67.0 | 75.4 | 66.6 | 56.0 | 65.3 |
| | msmarco-bert-co-condensor | 110M | 64.1 | 66.9 | 34.9 | 82.3 | 41.9 | 60.2 | 91.3 | 71.1 | 70.4 | 73.7 | 64.0 | 55.7 | 64.7 |
| | paraphrase-multilingual-MiniLM-L12-v2 | 118M | 71.5 | 69.2 | 35.1 | 79.8 | 42.3 | 60.5 | 87.0 | 65.5 | 66.9 | 71.5 | 60.1 | 56.1 | 63.8 |
| | sentence-t5-base | 110M | 75.8 | 85.1 | 44.9 | 76.5 | 51.4 | 77.3 | 90.3 | 63.3 | 69.7 | 72.3 | 68.2 | 62.7 | 69.8 |
| | text2vec-base-multilingual | 118M | 71.0 | 66.1 | 33.1 | 78.1 | 43.4 | 59.4 | 81.0 | 62.8 | 63.8 | 67.0 | 66.0 | 55.2 | 62.2 |
| | Angle_BERT | 109M | 77.9 | 76.0 | 37.2 | 75.5 | 45.2 | 68.8 | 85.4 | 64.5 | 66.3 | 70.6 | 67.1 | 57.6 | 66.0 |
| | gte-base | 109M | 74.2 | 91.8 | 49.0 | 85.1 | 48.6 | 86.0 | 93.0 | 72.0 | 71.5 | 76.4 | 71.6 | 57.0 | 73.0 |
| | ALL_862873 | 118M | 50.8 | 52.6 | 22.6 | 36.4 | 22.8 | 50.8 | 61.0 | 29.7 | 34.3 | 44.1 | 54.9 | 40.8 | 41.7 |
| MSE | Student-m | 109M | 76.6 | 89.1 | 44.7 | 87.2 | **60.8** | 88.0 | 95.7 | 81.6 | 77.7 | 82.2 | 67.3 | 60.5 | 76.0 |
| NLL | Student-m | 109M | 79.6 | 89.5 | 45.8 | **88.0** | 59.7 | 88.3 | **96.2** | **83.9** | **78.6** | **82.7** | 67.1 | 61.3 | **76.7** |

Table 14: Performance of our distilled models compared to models of similar sizes 200M to 420M parameters from the MTEB Benchmark on classification tasks.

| Task | Model | Size | Amazon Counterfactual | Amazon Polarity | Amazon Reviews | Banking77 | Emotion | Imdb | MTOPDomain | MTOPIntent | Massive Intent | Massive Scenario | Toxic Conversations | Tweet Sentiment Extraction | Avg. |
|---|---|---|---|---|---|---|---|---|---|---|---|---|---|---|---|
| MTEB | gte-multilingual-base | 305M | 76.0 | 80.7 | 43.6 | 85.4 | 48.0 | 74.9 | 92.5 | 72.6 | 72.1 | 76.3 | 71.0 | 57.6 | 70.9 |
| | bge-large-en-v1.5 | 335M | 75.8 | 92.4 | 48.2 | 87.8 | 51.5 | 92.8 | 94.6 | 79.5 | **77.6** | 80.5 | 70.9 | 59.9 | 76.0 |
| | GIST | 335M | 75.6 | 93.4 | 49.1 | **88.1** | 54.7 | 91.2 | **95.2** | 78.2 | 76.2 | 79.3 | **71.9** | 59.2 | 76.0 |
| | MUG-B-1.6 | 335M | 72.4 | 93.7 | **50.9** | 85.4 | 55.9 | **93.6** | 94.2 | 67.5 | 73.9 | 77.4 | 67.3 | 61.8 | 74.5 |
| | bilingual-embedding-base | 278M | 77.4 | 89.5 | 46.1 | 78.5 | 47.1 | 87.4 | 92.9 | 64.8 | 68.9 | 75.2 | 63.4 | 62.5 | 71.1 |
| | snowflake-arctic-embed-l | 334M | 74.8 | 78.4 | 36.7 | 80.1 | 46.5 | 72.9 | 92.6 | 65.8 | 71.1 | 64.7 | 64.7 | 56.7 | 67.1 |
| | UAE-Large-V1 | 335M | 75.5 | 92.8 | 48.3 | 87.7 | 51.8 | 92.8 | 94.0 | 76.9 | 76.5 | 79.8 | 71.1 | 59.8 | 75.6 |
| | embedder-100p | 278M | 67.1 | 70.4 | 33.2 | 82.7 | 43.5 | 67.3 | 91.8 | 74.7 | 71.8 | 77.8 | 67.5 | 55.6 | 67.0 |
| | instructor-large | 335M | **88.1** | 91.5 | 47.9 | 78.5 | 52.7 | 88.3 | 93.9 | 68.0 | 68.9 | 73.3 | 71.0 | **64.1** | 73.9 |
| | e5-large | 335M | 77.7 | 90.0 | 43.0 | 84.1 | 48.0 | 82.1 | 93.9 | 76.4 | 73.2 | 77.4 | 70.6 | 61.2 | 73.1 |
| | e5-large-v2 | 335M | 79.2 | **93.8** | 48.6 | 84.5 | 49.5 | 91.7 | 94.6 | 77.1 | 73.8 | 78.1 | 70.9 | 60.9 | 75.2 |
| | multilingual-e5-base | 278M | 77.4 | 91.8 | 47.5 | 73.5 | 45.7 | 84.3 | 90.9 | 61.6 | 65.7 | 71.6 | 64.3 | 62.8 | 69.8 |
| | sf_model_e5 | 335M | 70.8 | 91.8 | 48.9 | 84.6 | 54.9 | **93.1** | 93.6 | 66.0 | 73.5 | 77.4 | 71.2 | 61.5 | 74.0 |
| | jina-embedding-l-en-v1 | 335M | 68.9 | 69.1 | 31.4 | 85.3 | 45.8 | 66.4 | 92.8 | 76.1 | 72.7 | 77.1 | 69.1 | 58.2 | 67.8 |
| | ember-v1 | 335M | 76.1 | 92.0 | 47.9 | 87.9 | 52.0 | 92.8 | 94.6 | 79.3 | 77.4 | 80.5 | 71.4 | 60.0 | 76.0 |
| | mxbai-embed-2d-large-v1 | 335M | 74.8 | 93.3 | 46.2 | 86.7 | 49.3 | 90.4 | 93.1 | 73.2 | 73.9 | 78.2 | 71.5 | 59.2 | 74.1 |
| | mxbai-embed-large-v1 | 335M | 75.0 | **93.8** | 49.2 | 87.8 | 50.9 | 92.8 | 94.0 | 76.8 | 76.2 | 80.0 | 71.5 | 59.7 | 75.6 |
| | paraphrase-multilingual-mpnet-base-v2 | 278M | 75.8 | 76.4 | 38.5 | 81.1 | 45.8 | 64.6 | 89.2 | 68.7 | 69.3 | 75.3 | 71.0 | 59.0 | 67.9 |
| | gte-large | 335M | 72.6 | 92.5 | 49.1 | 86.1 | 47.9 | 88.5 | 93.5 | 73.2 | 72.6 | 76.8 | 70.6 | 56.6 | 73.3 |
| | b1ade-embed | 335M | 75.2 | 93.1 | 48.4 | **88.0** | 51.9 | 91.9 | 94.3 | 76.9 | 75.9 | 79.4 | 67.9 | 59.2 | 75.2 |
| MSE | Student-l | 335M | 77.3 | 84.5 | 43.4 | 86.0 | **60.0** | 82.7 | 95.1 | 79.8 | 76.3 | 81.3 | 65.8 | 60.2 | 74.4 |
| NLL | Student-l | 335M | 81.5 | 88.1 | 45.9 | 86.9 | **60.4** | 88.2 | **95.6** | **83.2** | 77.5 | **81.4** | 67.7 | 62.2 | **76.5** |

Table 15: Performance of our distilled models compared of models of similar sizes 16M to 30M parameters from the MTEB Benchmark on clustering tasks.

| Task
Model | | Size | Arxiv Clustering P2P | Arxiv Clustering S2S | Reddit Clustering P2P | Reddit Clustering | Stack Exchange Clustering P2P | Stack Exchange Clustering | Twenty Newsgroups Clustering | Avg. |
|---|---|---|---|---|---|---|---|---|---|---|
| MTEB | Bulbasaur | 17M | 40.3 | 31.1 | 51.4 | 45.9 | 30.7 | 52.2 | 39.4 | 41.6 |
| | Ivysaur | 23M | 46.4 | 35.4 | 56.0 | 47.5 | 33.6 | 53.9 | 40.8 | 44.8 |
| | Squirtle | 16M | 33.0 | 24.7 | 43.7 | 31.4 | 29.2 | 39.2 | 28.2 | 32.8 |
| | Venusaur | 16M | 31.8 | 21.1 | 44.1 | 26.7 | 27.5 | 32.8 | 26.1 | 30.0 |
| | Wartortle | 17M | 35.8 | 27.3 | 46.1 | 35.9 | 29.9 | 45.3 | 31.7 | 36.0 |
| | gte-micro | 17M | 35.2 | 31.1 | 47.9 | 45.6 | 30.1 | 52.6 | 40.8 | 40.5 |
| | gte-micro-v4 | 19M | 42.9 | 32.5 | 53.6 | 48.3 | 31.9 | 55.1 | 41.4 | 43.6 |
| | snowflake-arctic-embed-xs | 23M | 43.5 | 32.1 | 57.8 | 48.3 | 34.6 | 57.5 | 36.3 | 44.3 |
| | bge-micro | 17M | 44.6 | 34.5 | 54.5 | 45.3 | 34.7 | 53.1 | 39.4 | 43.7 |
| | bge-micro-v2 | 17M | 44.5 | 33.2 | 55.2 | 45.5 | 34.1 | 54.5 | 40.2 | 43.9 |
| | gte-tiny | 23M | **46.6** | 36.0 | 56.5 | 50.2 | **35.7** | 57.5 | 43.3 | 46.6 |
| | GIST-all-MiniLM-L6-v2 | 23M | 45.3 | 35.5 | 48.7 | 44.1 | 33.9 | 53.1 | 41.1 | 43.1 |
| | slx-v0.1 | 23M | 46.5 | 37.7 | 54.8 | 50.7 | 34.2 | 53.1 | **46.5** | 46.2 |
| | multi-qa-MiniLM-L6-cos-v1 | 23M | 37.8 | 27.7 | 51.0 | 46.3 | 33.4 | 48.1 | 40.8 | 40.7 |
| | all-MiniLM-L6-v2 | 23M | 46.5 | **37.9** | 54.8 | 50.7 | 34.3 | 53.1 | **46.5** | 46.3 |
| | rubert-tiny-turbo | 29M | 24.8 | 16.7 | 40.5 | 26.3 | 28.0 | 33.5 | 19.9 | 27.1 |
| MSE | Student-xs | 23M | 42.4 | 30.9 | 55.2 | 49.2 | 32.7 | 53.5 | 41.9 | 43.7 |
| NLL | Student-xs | 23M | 45.2 | 33.9 | **58.1** | **52.1** | 33.1 | **59.9** | 44.3 | **46.7** |

Table 16: Performance of our distilled models compared of models of similar sizes 30M to 50M parameters from the MTEB Benchmark on clustering tasks.

| Task
Model | | Size | Arxiv Clustering P2P | Arxiv Clustering S2S | Reddit Clustering P2P | Reddit Clustering | Stack Exchange Clustering P2P | Stack Exchange Clustering | Twenty Newsgroups Clustering | Avg. |
|---|---|---|---|---|---|---|---|---|---|---|
| MTEB | bge-small-en-v1.5 | 33M | 47.4 | 40.0 | 60.6 | 52.3 | 35.3 | 60.8 | 48.5 | 49.3 |
| | snowflake-arctic-embed-s | 33M | 44.9 | 35.9 | 60.5 | 50.5 | 34.0 | 60.7 | 38.3 | 46.4 |
| | bge-small-4096 | 35M | 43.9 | 29.6 | 54.3 | 43.7 | 33.3 | 51.8 | 36.6 | 41.9 |
| | GIST-small-Embedding-v0 | 33M | 47.6 | 39.9 | 60.6 | 55.5 | 36.2 | 61.9 | **50.0** | 50.2 |
| | NoInstruct-small-Embedding-v0 | 33M | 47.8 | 40.1 | 61.2 | 55.4 | **36.6** | 62.0 | 49.9 | 50.4 |
| | e5-small | 33M | 44.1 | 37.1 | 57.2 | 43.3 | 30.8 | 59.6 | 37.6 | 44.3 |
| | e5-small-v2 | 33M | 42.1 | 34.8 | 59.7 | 45.7 | 32.0 | 58.5 | 41.1 | 44.8 |
| | jina-embedding-s-en-v1 | 35M | 34.2 | 24.0 | 49.9 | 38.0 | 31.5 | 46.4 | 34.4 | 36.9 |
| | jina-embeddings-v2-small-en | 33M | 44.0 | 35.2 | 57.1 | 49.3 | 34.4 | 55.4 | 41.6 | 45.3 |
| | all-MiniLM-L12-v2 | 33M | 46.1 | 37.5 | 54.8 | 51.2 | 33.1 | 53.0 | 47.5 | 46.2 |
| | gte-small | 33M | **47.9** | **40.3** | **61.4** | **55.6** | 36.3 | **62.6** | 50.0 | **50.6** |
| MSE | Student-s | 33M | 43.1 | 33.3 | 57.1 | 50.8 | 32.3 | 55.7 | 42.8 | 45.0 |
| NLL | Student-s | 33M | 45.9 | 35.2 | 60.3 | 51.9 | 32.3 | 61.5 | 45.1 | 47.4 |

Table 17: Performance of our distilled models compared of models of similar sizes 100M to 120M parameters from the MTEB Benchmark on clustering tasks.

| | Task / Model | Size | Arxiv Clustering P2P | Arxiv Clustering S2S | Reddit Clustering P2P | Reddit Clustering | Stack Exchange Clustering P2P | Stack Exchange Clustering | Twenty Newsgroups Clustering | Avg. |
|---|---|---|---|---|---|---|---|---|---|---|
| MTEB | bge-base-en-v1.5 | 109M | **48.8** | 42.8 | 62.7 | 56.6 | 35.2 | 66.1 | 50.8 | 51.8 |
| | bilingual-embedding-small | 118M | 41.8 | 31.6 | 58.4 | 47.4 | 33.6 | 52.5 | 40.5 | 43.7 |
| | multilingual-e5-small | 118M | 39.2 | 30.8 | 59.0 | 39.1 | 32.1 | 53.5 | 33.2 | 41.0 |
| | snowflake-arctic-embed-m | 109M | 47.2 | 37.4 | 62.8 | 47.5 | **39.4** | 59.5 | 37.7 | 47.4 |
| | snowflake-arctic-embed-m-v1.5 | 109M | 45.0 | 34.1 | 61.8 | 51.9 | 33.8 | 61.2 | 38.1 | 46.6 |
| | GIST-Embedding-v0 | 109M | 48.3 | 42.7 | 62.4 | 59.1 | 35.6 | _66.1_ | _52.2_ | _52.4_ |
| | ml-nlp-elser.html | 110M | 35.3 | 23.2 | 51.9 | 38.7 | 28.7 | 42.7 | 27.8 | 35.5 |
| | e5-base-4k | 112M | 46.1 | 39.7 | **63.4** | 56.2 | 32.5 | 65.2 | 48.2 | 50.2 |
| | instructor-base | 110M | 39.7 | 29.2 | 63.2 | _59.3_ | 35.3 | 65.0 | 51.3 | 49.0 |
| | bert-base-uncased | 110M | 35.2 | 27.5 | 43.3 | 27.2 | 26.6 | 43.6 | 23.4 | 32.4 |
| | e5-base | 109M | 44.6 | 40.5 | 62.2 | 48.2 | 32.6 | 63.9 | 42.6 | 47.8 |
| | e5-base-v2 | 110M | 46.1 | 39.7 | _63.2_ | 56.5 | 33.0 | 64.6 | 49.9 | 50.4 |
| | jina-embedding-b-en-v1 | 110M | 39.2 | 29.1 | 52.5 | 42.9 | 31.4 | 48.1 | 38.1 | 40.2 |
| | contriever-base-msmarco | 110M | 42.6 | 32.3 | 57.6 | 54.9 | 32.2 | 63.1 | 46.8 | 47.1 |
| | sup-simcse-bert-base-uncased | 110M | 35.2 | 27.5 | 47.7 | 40.2 | 29.4 | 47.5 | 34.9 | 37.5 |
| | unsup-simcse-bert-base-uncased | 110M | 32.6 | 24.7 | 45.1 | 32.2 | 28.5 | 43.1 | 23.2 | 32.8 |
| | all-mpnet-base-v2 | 110M | 48.4 | 39.7 | 56.8 | 54.8 | 34.3 | 53.8 | 49.7 | 48.2 |
| | allenai-specter | 110M | 44.8 | 35.3 | 35.1 | 24.1 | 31.5 | 39.0 | 24.2 | 33.4 |
| | gtr-t5-base | 110M | 35.5 | 27.2 | 58.5 | 56.1 | 33.0 | 64.2 | 46.7 | 45.9 |
| | msmarco-bert-co-condensor | 110M | 36.9 | 29.0 | 53.5 | 48.0 | 30.5 | 59.5 | 38.7 | 42.3 |
| | paraphrase-multilingual-MiniLM-L12-v2 | 118M | 38.3 | 31.6 | 50.1 | 42.6 | 31.7 | 49.3 | 40.0 | 40.5 |
| | sentence-t5-base | 110M | 39.3 | 27.3 | 59.7 | 52.9 | 35.7 | 63.1 | 48.1 | 46.6 |
| | text2vec-base-multilingual | 118M | 32.3 | 25.5 | 43.3 | 31.2 | 30.6 | 34.4 | 31.6 | 32.7 |
| | Angle_BERT | 109M | 35.3 | 27.7 | 46.0 | 40.3 | 28.9 | 48.3 | 33.1 | 37.1 |
| | gte-base | 109M | _48.6_ | **43.0** | 62.6 | _59.3_ | _36.0_ | **66.6** | **52.3** | **52.6** |
| | ALL_862873 | 118M | 14.8 | 12.2 | 27.1 | 18.4 | 27.3 | 23.7 | 20.2 | 20.5 |
| MSE | Student-m | 109M | 46.5 | 37.1 | 60.4 | 54.5 | 33.4 | 62.0 | 46.1 | 48.6 |
| NLL | Student-m | 109M | 47.7 | 38.7 | 61.5 | 56.3 | 33.8 | 64.7 | 46.6 | 49.9 |

Table 18: Performance of our distilled models compared of models of similar sizes 16M to 30M parameters from the MTEB Benchmark on STS tasks.

| | Task / Model | Size | BIOSSES | SICK-R | STS12 | STS13 | STS14 | STS15 | STS16 | STS17 | STS22 | STSBenchmark | Avg. |
|---|---|---|---|---|---|---|---|---|---|---|---|---|---|
| MTEB | Bulbasaur | 17M | 85.0 | 76.0 | 69.5 | 81.0 | 77.1 | 85.4 | 82.3 | 88.0 | 64.1 | 83.3 | 79.2 |
| | Ivysaur | 23M | **87.3** | 75.6 | 68.6 | 80.5 | 77.6 | 86.2 | 82.8 | **88.6** | _67.4_ | 84.2 | 79.9 |
| | Squirtle | 16M | 71.8 | 77.3 | 70.2 | 78.4 | 74.8 | 82.0 | 78.3 | 85.8 | 61.2 | 79.2 | 75.9 |
| | Venusaur | 16M | 77.6 | 74.7 | 54.4 | 74.2 | 70.0 | 75.7 | 73.7 | 84.8 | 62.6 | 76.7 | 72.4 |
| | Wartortle | 17M | 80.8 | 78.2 | **75.2** | 79.3 | 76.6 | 84.7 | 81.4 | 86.6 | 63.4 | 81.8 | 78.8 |
| | snowflake-arctic-embed-xs | 23M | 84.0 | 69.3 | 65.9 | 77.9 | 72.8 | 83.5 | 80.6 | 84.5 | 66.3 | 79.2 | 76.4 |
| | bge-micro | 17M | 83.4 | 72.4 | 71.9 | 80.9 | 76.6 | 84.9 | 80.7 | 85.6 | 65.9 | 81.3 | 78.4 |
| | bge-micro-v2 | 17M | 82.9 | 73.6 | 71.9 | 79.8 | 76.9 | 84.8 | 81.9 | 86.8 | 65.4 | 82.5 | 78.7 |
| | gte-tiny | 23M | _86.6_ | 75.8 | 72.6 | _82.4_ | _78.0_ | _86.5_ | **83.3** | 88.3 | 66.7 | _84.4_ | _80.5_ |
| | GIST-all-MiniLM-L6-v2 | 23M | 81.3 | _79.1_ | _75.0_ | **83.3** | **78.6** | **87.0** | _83.0_ | 87.4 | **68.1** | _84.4_ | **80.7** |
| | multi-qa-MiniLM-L6-cos-v1 | 23M | 79.8 | 70.0 | 64.4 | 76.4 | 69.3 | 80.2 | 79.6 | 81.2 | 65.5 | 76.0 | 74.2 |
| | all-MiniLM-L6-v2 | 23M | 81.6 | 77.6 | 72.4 | 80.6 | 75.6 | 85.4 | 79.0 | 87.6 | 67.2 | 82.0 | 78.9 |
| MSE | Student-xs | 23M | 76.8 | **79.2** | 72.2 | 80.3 | 75.9 | 85.0 | 83.0 | 87.1 | 66.4 | 82.9 | 78.9 |
| NLL | Student-xs | 23M | 78.8 | 77.8 | 71.6 | 80.2 | 77.0 | 85.8 | 82.8 | **89.3** | 65.8 | 83.5 | 79.3 |

Table 19: Performance of our distilled models compared of models of similar sizes 30M to 50M parameters from the MTEB Benchmark on STS tasks.

| | Task
Model | Size | BIOSSES | SICK-R | STS12 | STS13 | STS14 | STS15 | STS16 | STS17 | STS22 | STSBenchmark | Avg. |
|---|---|---|---|---|---|---|---|---|---|---|---|---|---|
| MTEB | bge-small-en-v1.5 | 33M | 83.8 | 79.4 | **77.4** | 83.0 | 81.8 | 87.3 | 84.9 | 87.2 | 65.3 | 85.9 | 81.6 |
| | snowflake-arctic-embed-s | 33M | 86.3 | 69.7 | 68.8 | 79.6 | 75.6 | 84.6 | 82.4 | 86.7 | **69.5** | 81.2 | 78.4 |
| | bge-small-4096 | 35M | 81.6 | 74.2 | 72.2 | 80.5 | 76.2 | 85.2 | 81.9 | 86.6 | 65.5 | 81.9 | 78.6 |
| | GIST-small-Embedding-v0 | 33M | 87.0 | **80.5** | 75.6 | **86.3** | **82.3** | **88.7** | **85.3** | **89.0** | 68.5 | **87.1** | **83.0** |
| | NoInstruct-small-Embedding-v0 | 33M | 87.2 | 80.3 | 75.8 | 86.1 | 82.3 | **88.9** | 85.2 | 88.7 | 68.5 | 87.0 | 83.0 |
| | e5-small | 33M | 84.2 | 78.9 | 75.2 | 81.8 | 78.5 | 87.5 | 84.6 | 87.9 | 63.8 | 86.4 | 80.9 |
| | e5-small-v2 | 33M | 79.4 | 78.5 | 76.2 | 82.4 | 79.0 | 87.8 | 83.8 | 87.7 | 63.1 | 86.0 | 80.4 |
| | jina-embedding-s-en-v1 | 35M | 83.0 | 76.3 | 74.3 | 78.5 | 73.8 | 83.7 | 80.0 | 87.5 | 64.2 | 79.2 | 78.1 |
| | jina-embeddings-v2-small-en | 33M | 80.5 | 76.7 | 73.7 | 83.3 | 79.2 | 87.3 | 83.6 | 88.2 | 63.5 | 84.0 | 80.0 |
| | all-MiniLM-L12-v2 | 33M | 83.6 | 79.3 | 73.1 | 82.1 | 76.7 | 85.6 | 80.2 | 88.6 | 65.7 | 83.1 | 79.8 |
| | gte-small | 33M | **88.2** | 77.9 | 75.1 | 85.1 | 81.0 | 88.3 | 83.9 | 87.6 | 68.0 | 85.6 | 82.1 |
| MSE | Student-s | 33M | 78.9 | 79.5 | 70.6 | 79.7 | 75.4 | 84.1 | 81.8 | 86.7 | 66.6 | 83.1 | 78.6 |
| NLL | Student-s | 33M | 81.5 | 79.3 | 73.0 | 81.4 | 78.2 | 86.3 | 84.2 | **90.0** | 66.0 | 84.8 | 80.5 |

Table 20: Performance of our distilled models compared of models of similar sizes 100M to 120M parameters from the MTEB Benchmark on STS tasks.

| | Task
Model | Size | BIOSSES | SICK-R | STS12 | STS13 | STS14 | STS15 | STS16 | STS17 | STS22 | STSBenchmark | Avg. |
|---|---|---|---|---|---|---|---|---|---|---|---|---|---|
| MTEB | bge-base-en-v1.5 | 109M | 86.9 | 80.3 | 78.0 | 84.2 | 82.3 | 88.0 | _85.5_ | 86.4 | 66.0 | 86.4 | _82.4_ |
| | bilingual-embedding-small | 118M | 84.0 | 74.7 | _79.4_ | 85.3 | _83.9_ | 88.5 | 84.4 | 85.8 | 67.2 | 86.1 | 81.9 |
| | multilingual-e5-small | 118M | 82.3 | 77.5 | 76.6 | 77.0 | 75.5 | 87.1 | 83.6 | 86.4 | 60.9 | 84.0 | 79.1 |
| | snowflake-arctic-embed-m | 109M | 86.6 | 69.1 | 67.0 | 79.1 | 68.5 | 79.9 | 78.7 | 81.5 | 65.8 | 74.1 | 75.0 |
| | snowflake-arctic-embed-m-v1.5 | 109M | 86.4 | 69.9 | 61.8 | 82.7 | 69.0 | 75.5 | 77.3 | 75.0 | _69.1_ | 69.7 | 73.6 |
| | GIST-Embedding-v0 | 109M | **88.0** | **81.3** | 76.2 | **87.8** | 83.4 | **89.4** | 85.3 | 88.6 | 67.8 | **87.3** | **83.5** |
| | ml-nlp-elser.html | 110M | 83.8 | 68.8 | 64.8 | 80.1 | 75.0 | 83.7 | 80.5 | 85.7 | 67.5 | 79.5 | 76.9 |
| | e5-base-4k | 112M | 81.4 | 78.3 | 75.8 | 83.6 | 80.0 | _88.8_ | 84.5 | 87.6 | 64.1 | _86.5_ | 81.0 |
| | instructor-base | 110M | 82.3 | 80.3 | 77.0 | _86.6_ | 81.3 | 88.2 | 84.9 | 89.5 | 66.5 | 86.4 | 82.3 |
| | bert-base-uncased | 110M | 54.7 | 58.6 | 30.9 | 59.9 | 47.7 | 60.3 | 63.7 | 64.1 | 56.4 | 47.3 | 54.4 |
| | e5-base | 109M | 85.1 | 79.7 | 74.2 | 83.3 | 78.5 | 88.3 | 84.2 | 87.2 | 62.9 | 86.2 | 81.0 |
| | e5-base-v2 | 110M | 81.4 | 78.3 | 75.8 | 83.6 | 80.0 | _88.8_ | 84.5 | 87.6 | 64.1 | _86.5_ | 81.0 |
| | jina-embedding-b-en-v1 | 110M | 83.6 | 79.1 | 75.1 | 80.9 | 76.1 | 85.5 | 81.2 | 89.0 | 66.2 | 82.6 | 79.9 |
| | contriever-base-msmarco | 110M | 83.3 | 70.2 | 64.3 | 80.0 | 74.5 | 83.3 | 79.7 | 86.3 | 64.6 | 78.8 | 76.5 |
| | sup-simcse-bert-base-uncased | 110M | 68.4 | 80.8 | 75.3 | 84.7 | 80.2 | 85.4 | 80.8 | 89.4 | 62.0 | 84.2 | 79.1 |
| | unsup-simcse-bert-base-uncased | 110M | 72.3 | 72.2 | 66.0 | 81.5 | 73.6 | 79.7 | 78.1 | 83.6 | 59.6 | 76.5 | 74.3 |
| | all-mpnet-base-v2 | 110M | 80.4 | 80.6 | 72.6 | 83.5 | 78.0 | 85.7 | 80.0 | **90.6** | _68.0_ | 83.4 | 80.3 |
| | allenai-specter | 110M | 65.0 | 56.4 | 62.5 | 58.7 | 54.9 | 62.5 | 64.3 | 69.6 | 55.1 | 61.3 | 61.0 |
| | gtr-t5-base | 110M | 79.0 | 71.5 | 68.6 | 79.1 | 74.6 | 84.8 | 81.6 | 85.8 | 66.2 | 79.6 | 77.1 |
| | msmarco-bert-co-condensor | 110M | 77.3 | 72.0 | 68.2 | 80.4 | 74.0 | 82.6 | 79.8 | 85.9 | 67.5 | 77.0 | 76.5 |
| | paraphrase-multilingual-MiniLM-L12-v2 | 118M | 74.2 | 79.6 | 76.0 | 80.7 | 78.8 | 85.8 | 81.0 | 86.9 | 62.1 | 84.4 | 79.0 |
| | sentence-t5-base | 110M | 75.9 | 80.2 | 78.0 | 85.8 | 82.2 | 87.5 | 84.0 | 89.6 | 62.7 | 85.5 | 81.1 |
| | text2vec-base-multilingual | 118M | 66.2 | 80.0 | **80.9** | 82.9 | **87.4** | 88.3 | 81.6 | 85.8 | 63.0 | 86.5 | 80.2 |
| | gte-base | 109M | _87.6_ | 78.9 | 75.7 | 85.7 | 81.5 | 88.8 | 83.8 | 87.9 | 67.3 | 85.7 | 82.3 |
| | ALL_862873 | 118M | 21.3 | 48.5 | 55.6 | 18.4 | 28.8 | 29.2 | 39.0 | 61.2 | 44.5 | 44.4 | 39.1 |
| MSE | Student-m | 109M | 83.4 | _80.9_ | 74.5 | 82.8 | 79.0 | 86.6 | 85.2 | 88.4 | 66.4 | 85.2 | 81.2 |
| NLL | Student-m | 109M | 85.2 | 80.2 | 75.2 | 83.4 | 80.4 | 88.3 | **86.0** | 89.9 | 66.2 | 86.4 | 82.1 |

Table 21: Performance of our distilled models compared of models of similar sizes 200M to 400M parameters from the MTEB Benchmark on STS tasks.

| Task | Model | Size | BIOSSES | SICK-R | STS12 | STS13 | STS14 | STS15 | STS16 | STS17 | STS22 | STSBenchmark | Avg. |
|------|-------|------|---------|--------|-------|-------|-------|-------|-------|-------|-------|--------------|------|
| MTEB | gte-multilingual-base | 305M | 81.2 | 79.3 | 77.5 | 85.5 | 81.7 | 89.0 | 84.3 | 88.9 | 67.2 | 86.5 | 82.1 |
| | bge-large-en-v1.5 | 335M | 84.7 | 81.7 | 79.0 | 86.4 | 82.8 | 88.0 | 86.5 | 87.5 | 67.0 | 87.5 | 83.1 |
| | MUG-B-1.6 | 335M | 88.4 | **83.0** | 79.2 | 89.4 | 84.8 | 89.5 | 86.7 | 89.6 | **70.3** | 89.0 | 85.0 |
| | bilingual-embedding-base | 278M | 87.1 | 79.5 | **79.6** | 84.7 | 83.9 | 89.9 | 84.9 | 88.7 | 64.3 | 87.4 | 83.0 |
| | snowflake-arctic-embed-l | 334M | 86.3 | 69.3 | 67.8 | 77.5 | 69.8 | 80.2 | 77.9 | 82.3 | 68.0 | 75.7 | 75.5 |
| | UAE-Large-V1 | 335M | 86.1 | 82.6 | 79.1 | 89.6 | 85.0 | 89.5 | 86.6 | 89.0 | 68.8 | 89.1 | 84.5 |
| | GIST-large-Embedding-v0 | 335M | **89.2** | 82.8 | 77.1 | 89.3 | 83.8 | 89.7 | 86.4 | 89.7 | 69.6 | 88.3 | 84.6 |
| | embedder-100p | 278M | 75.3 | 80.9 | 77.0 | 82.6 | 77.8 | 85.9 | 80.7 | 89.0 | 68.3 | 84.2 | 80.2 |
| | instructor-large | 335M | 84.4 | 81.3 | 76.3 | 88.2 | 81.9 | 89.0 | 85.5 | **90.3** | 67.7 | 86.9 | 83.1 |
| | e5-large | 335M | 84.7 | 80.5 | 75.9 | 85.2 | 80.5 | 88.8 | 85.3 | 89.4 | 63.0 | 87.2 | 82.1 |
| | e5-large-v2 | 335M | 83.6 | 79.3 | 77.0 | 84.1 | 80.5 | 89.8 | 85.5 | 89.0 | 64.1 | 87.7 | 82.1 |
| | multilingual-e5-base | 278M | 85.0 | 78.5 | 76.7 | 78.0 | 76.6 | 88.2 | 84.3 | 87.8 | 62.3 | 85.6 | 80.3 |
| | sf_model_e5 | 335M | 86.8 | 82.3 | 77.6 | 88.0 | 83.8 | 88.5 | 86.5 | 88.7 | 68.0 | 88.3 | 83.8 |
| | jina-embedding-l-en-v1 | 335M | 84.4 | 79.2 | 74.5 | 83.2 | 78.1 | 86.9 | 83.7 | 90.2 | 64.9 | 84.6 | 81.0 |
| | ember-v1 | 335M | 85.8 | 81.8 | 78.5 | 86.6 | 83.1 | 88.4 | 86.8 | 87.9 | 66.8 | 87.8 | 83.3 |
| | mxbai-embed-2d-large-v1 | 335M | 88.1 | 82.0 | 78.8 | **90.4** | 85.5 | **90.0** | **87.4** | 88.8 | 68.8 | **89.2** | 84.9 |
| | mxbai-embed-large-v1 | 335M | 88.4 | 82.9 | 78.8 | 90.3 | **85.5** | 89.6 | 86.6 | 89.5 | 69.3 | 89.1 | **85.0** |
| | paraphrase-multilingual-mpnet-base-v2 | 278M | 76.3 | 79.6 | 77.9 | 85.1 | 80.8 | 87.5 | 83.2 | 87.0 | 63.5 | 86.8 | 80.8 |
| | gte-large | 335M | 88.7 | 79.8 | 76.8 | 88.1 | 82.7 | 88.9 | 84.2 | 88.5 | 69.7 | 86.1 | 83.3 |
| | b1ade-embed | 335M | 89.2 | 82.8 | 78.7 | 90.0 | 85.0 | 89.8 | 86.7 | 89.8 | 69.7 | 88.8 | **85.0** |
| MSE | Student-l | 335M | 79.1 | 80.6 | 73.7 | 82.1 | 78.1 | 87.4 | 84.2 | 89.1 | 67.0 | 85.3 | 80.7 |
| NLL | Student-l | 335M | 83.8 | 79.5 | 74.4 | 83.0 | 79.6 | 88.0 | 85.2 | 90.1 | 65.3 | 86.2 | 81.5 |

Table 22: Head-to-head comparison on selected MTEB classification tasks, with large embedders (over x5 times the number of parameters).

| | Model | Size | AmazonCtf | Banking77 | IMDB | MTOP Dom. | Massive Int. | Massive Scen. | Toxic Conv. | Tweet Sent. | Avg. |
|---|-------|------|-----------|-----------|------|-----------|--------------|---------------|-------------|-------------|------|
| | Qwen3-Embedding-4B | 4.0B | 93.7 | 86.3 | 97.2 | 97.8 | 85.0 | 88.8 | 91.4 | 78.4 | 89.8 |
| | stella_en_1.5B_v5 | 1.5B | 94.1 | 89.8 | 96.7 | 98.7 | 84.5 | 89.7 | 86.8 | 74.8 | 89.4 |
| | jasper_en_vision_language_v1 | 1.0B | 93.8 | 87.2 | 97.0 | 99.2 | 85.3 | 91.2 | 91.3 | 77.2 | 90.3 |
| | Qwen3-Embedding-0.6B | 595M | 91.5 | 81.0 | 95.4 | 96.0 | 80.4 | 83.6 | 82.1 | 76.0 | 85.8 |
| | jina-embeddings-v3 | 572M | 90.9 | 84.1 | 91.9 | – | 75.2 | 84.1 | 91.3 | 71.4 | 84.1 |
| | snowflake-arctic-embed-l-v2.0 | 568M | 65.6 | 81.8 | 72.8 | 93.5 | 71.5 | 76.2 | 65.9 | 59.6 | 73.4 |
| | KaLM-embed-mini-instr-v2 | 494M | 95.3 | 89.5 | 95.2 | 98.9 | 77.8 | 86.0 | 89.3 | 78.6 | 88.8 |
| | KaLM-embed-mini-instr-v1 | 494M | 81.5 | 84.9 | 95.0 | 92.2 | 69.8 | 74.2 | 89.0 | 76.5 | 82.9 |
| | KaLM-embed-mini-v1 | 494M | 76.4 | 79.2 | 91.6 | 92.5 | 70.9 | 76.1 | 70.8 | 62.7 | 77.5 |
| | stella_en_400M_v5 | 435M | 94.3 | 89.3 | 96.5 | 98.3 | 80.5 | 89.6 | 84.0 | 73.6 | 88.2 |
| NLL | Student-m-nll | 109M | 79.6 | 88.0 | 88.3 | 96.2 | 78.6 | 82.7 | 67.1 | 61.3 | 80.2 |
| | Student-s-nll | 32M | 77.3 | 86.7 | 88.3 | 95.5 | 76.7 | 80.7 | 66.1 | 60.6 | 79.0 |

# D  Vision

## D.1  Model architecture

The models we used for vision as teachers and student are presented in Tab. 23, including the number of parameters of each of them.

## D.2  Training Set

Tab. 24 presents the statistics, *i.e.* the number of training and testing samples, of the datasets we used for vision.

## D.3  Vision Details

**Data processing details:**  We use the official train sets of the datasets for the knowledge distillation part. We split the official training part, if there are no official validation sets, to train and validation set with 80 and 20 percents of the data, consequently. For the augmentation we used color jitter with brightness, contrast, saturation and hue equal to 0.2, and random horizontal flip (except for the SVHN dataset).

**Distillation details:**  For training the distillation, we extract the embeddings of the train set of each dataset, for each teacher and divide the embeddings to 80 train set and 20 percent validation set. For the optimizer we use Adam, with learning rate of 0.001, a batch size of 128, trained for 50 epochs.

**Down-stream task fine-tuning:**  For fine-tuning of down-stream tasks, we add a classifier on the frozen embedders. We again use Adam optimizer for the fine-tuning of downstream tasks. We perform hyperparameter tuning using grid search to optimize the performance of our models. Our search space includes the learning rate with values (1e-2, 1e-3), the number of fully connected layer units with values (0, 128), and the type of normalization after the fully connected layer, considering (no optimization, batch normalization, layer normalization). The models are trained for a maximum of 1000 epochs with a batch size of 128, but we apply early stopping with a patience of 20 to prevent over-fitting and reduce unnecessary computation.

## D.4  Complementary Results

Tab. 25 shows the detailed results of the Vision Transformer teachers and students. The best among the students are shown with an underline, showing that on average and most of the cases our method improves the baseline. In addition to the main results, we added additional experiments to answer further informative question:

Table 23: Number of parameters for each model (in million parameters)

| Model | # Parameters |
|---|---|
| Swin (Liu et al., 2021b) | 87.77M |
| DINOv2 (Oquab et al., 2023) | 86.58M |
| ViT (Dosovitskiy et al., 2021) | 86.57M |
| BEiT (Bao et al., 2022) | 86.53M |
| PVTv2 (Wang et al., 2022c) | 3.67M |
| WideResNet (Zagoruyko & Komodakis, 2017) | 68.88M |
| DenseNet (Huang et al., 2017) | 28.68M |
| ResNext (Xie et al., 2017) | 25.03M |
| ResNet18 (He et al., 2016) | 11.69M |
| GoogLeNet (Szegedy et al., 2015) | 6.62M |
| MNASNet (Tan et al., 2019) | 4.38M |
| MobileNet (Sandler et al., 2018) | 3.50M |
| ShuffleNet (Ma et al., 2018) | 2.28M |
| SqueezeNet (Iandola et al., 2016) | 1.25M |

Table 24: Number of classes, training, validation (if any) and testing samples in each vision dataset

| Dataset | classes | training samples | validation samples | test samples |
|---|---|---|---|---|
| CIFAR10 (Krizhevsky et al., 2009) | 10 | 50000 | - | 10000 |
| STL10 (Coates et al., 2011) | 10 | 5000 | - | 8000 |
| SVHN (Netzer et al., 2011) | 10 | 73257 | - | 26032 |
| CUB (Welinder et al., 2010) | 200 | 5,994 | - | 5,794 |
| DTD (Cimpoi et al., 2014) | 47 | 1880 | 1880 | 1880 |
| FGVCAircraft (Maji et al., 2013) | 100 | 3334 | 3333 | 3333 |
| Oxford Pets (Parkhi et al., 2012) | 37 | 3680 | - | 8041 |
| Food101 (Bossard et al., 2014) | 101 | 750 | - | 250 |
| Stanford Cars (Krause et al., 2013) | 196 | 8144 | - | 8041 |

Table 25: Comparison of Vision Transformer teachers, CNN baselines and the ViT student, with their corresponding parameter size, with the underline showing the best students.

| Method | Model | # Parameters | CIFAR10 | DTD | STL10 | SVHN | FGVCAircraft | CUB |
|---|---|---|---|---|---|---|---|---|
| NoKD | Swin | 87.77 | 97.67 | 76.33 | **99.60** | 64.42 | 52.45 | 87.11 |
| | ViT | 86.57 | 96.90 | 71.65 | 99.40 | 54.97 | 41.71 | 82.67 |
| | DINOv2 | 86.58 | **98.57** | **83.30** | 99.45 | 63.01 | **79.40** | **89.02** |
| | BEiT | 86.53 | 97.89 | 77.34 | **99.60** | 66.61 | 55.45 | 39.52 |
| | PVTv2 | 3.67 | 89.27 | 65.05 | 95.80 | 62.03 | 38.58 | 68.97 |
| | wide resnet | 68.88 | 85.65 | 65.37 | 95.85 | 57.77 | 30.82 | 60.55 |
| | densenet | 28.68 | 87.49 | 67.93 | 97.11 | 66.91 | 46.84 | 68.62 |
| | resnet18 | 11.69 | 83.22 | 61.54 | 92.98 | 51.01 | 36.09 | 59.89 |
| | googlenet | 6.62 | 82.07 | 66.38 | 93.95 | 55.90 | 35.85 | 59.09 |
| CompRess | PVTv2 | 3.67 | 94.6 | 52.7 | 93.5 | 61.9 | 32.7 | 48.8 |
| MSE | PVTv2 | 3.67 | 96.1 | 65.1 | 96.4 | 70.3 | 34.4 | 67.7 |
| Cosine | PVTv2 | 3.67 | 95.89 | 65.4 | 96.7 | 70.7 | 35.9 | 67.1 |
| RKD | PVTv2 | 3.67 | 87.64 | 52.23 | 89.63 | 61.66 | 30.54 | 47.85 |
| CC grbf | PVTv2 | 3.67 | 84.07 | 61.86 | 93.03 | 59.96 | 33.48 | 57.55 |
| CC bilinear | PVTv2 | 3.67 | 92.95 | 61.22 | 95.42 | 63.71 | 35.16 | 64.70 |
| NLL | PVTv2 | 3.67 | 94.76 | 65.85 | 96.45 | **76.91** | 48.13 | 69.37 |

**How will our method work in vision for unseen datasets?** Tab. 26 shows the accuracy of our student compared to various distillation baselines: MSE distillation, Cosine distillation, Correlation Congruence (CC rbf and CC dot) Peng et al. (2019), CompRess Abbasi Koohpayegani et al. (2020) and relational KD Park et al. (2019b).

for three unseen datasets. As we can see, our method improved the baselines considerably for unseen datasets.

**How our method works for a setting with diverse teachers specialized in different task, and if it will be able to avoid conflicts?** We evaluated the student model's classification performance using three specialized vision teachers: ViT (classification), DETR ( (Carion et al., 2020) , object detection),

Table 26: Comparison of ViT student of our method (NLL), and various distillation baselines for the unseen datasets.

| Method | Oxford Pets | Food101 | Stanford Cars |
|---|---|---|---|
| CompRess | 70.23 | 45.48 | 19.43 |
| MSE | 85.58 | 58.04 | 31.96 |
| Cosine | 84.38 | 56.37 | 30.92 |
| RKD | 69.99 | 43.48 | 18.24 |
| CC rbf | 85.09 | 58.47 | 30.08 |
| CC dot | 67.42 | 45.93 | 20.88 |
| NLL | **87.46** | **62.62** | **41.29** |

and SegFormer ( (Xie et al., 2021), segmentation). We also included DINOv2, a general-purpose embedding model known for strong performance across multiple benchmarks. As shown in Tab. 27, adding DETR or SegFormer alongside ViT did not significantly improve or degrade classification performance compared to using ViT alone. This suggests that while task-specific teachers may offer limited benefit outside their domain, they do not negatively impact the student's learning.

To further validate this, we incorporated DINOv2 into the teacher set ( Tab. 28). This addition improved overall performance, while the inclusion of DETR and SegFormer continued to have minimal effect, confirming that our earlier observations hold even in a more competitive setting with a strong general-purpose teacher. These results are consistent with Sec. 5.2 and Figure C.1.3, where we observe that adding teachers typically boosts student performance. In molecular and text domains, where all teachers are general-purpose embedders, improvements are more uniform. However, in vision tasks, specialized teachers contribute gains primarily in their area of expertise, yet without harming performance elsewhere. Overall, these findings suggest that our method can effectively integrate knowledge from both specialized and generalist teachers without conflict.

Table 27: Performance of different teacher combinations across datasets (accuracy %).

| Teachers | CIFAR-10 | DTD | STL-10 | SVHN | FGVC | CUB | Average |
|---|---|---|---|---|---|---|---|
| ViT + Segformer + DETR | 94.03 | 63.62 | 95.86 | 65.63 | 38.79 | 67.67 | **70.93** |
| ViT + Segformer | 94.23 | 63.24 | **95.91** | **65.79** | 38.31 | 67.35 | 70.81 |
| ViT + DETR | **94.71** | 61.28 | 95.80 | 64.14 | 37.89 | 65.90 | 69.95 |
| ViT | 94.69 | 61.70 | 95.75 | 64.13 | **39.42** | **69.23** | 70.82 |
| DETR + Segformer | 87.87 | **63.72** | 94.81 | 54.71 | 37.89 | 62.43 | 66.91 |

Table 28: Comparison of ViT-based teacher combinations including DINO on multiple datasets (accuracy %). Bolded values indicate best per column.

| Teachers | CIFAR-10 | DTD | STL-10 | SVHN | FGVC | CUB | Average |
|---|---|---|---|---|---|---|---|
| ViT + Segformer + DETR + DINO | **95.39** | 64.31 | 96.14 | **72.88** | 50.38 | 69.69 | **74.80** |
| ViT + DINO | 95.83 | **61.92** | **96.06** | 73.60 | 50.59 | **69.21** | 74.54 |

As another additional experiment, we use CNN based teachers for resnet18, for different relevant datasets. Tab. 29 shows the performance improvements, and the effectiveness of using our distillation method, compared to other.

Table 29: Comparison of the performance with CNN-based teacher (accuracy %). Bolded values indicate best per column.

| Method | Model | CIFAR10 | FMNIST | MNIST | STL10 | SVHN | QMNIST | KMNIST | CelebA |
|---|---|---|---|---|---|---|---|---|---|
| NoKD | resnet18 | 81.89 | 86.94 | 96.6 | 92.98 | 51.01 | 96.89 | 80.43 | 90.82 |
| | squeezenet | 79.23 | 86.65 | 97.51 | 85.82 | 47.77 | 97.59 | 84.05 | 61.35 |
| | densenet | 87.49 | 88.69 | 96.80 | **97.11** | 66.91 | 97.72 | 86.33 | 93.98 |
| | googlenet | 81.94 | 86.38 | 96.71 | 93.95 | 55.9 | 97.2 | 79.27 | 92.93 |
| | shufflenet | 81.61 | 87.57 | 95.77 | 71.51 | 49.08 | 95.96 | 76.97 | 92.42 |
| | mobilenet | 81.67 | 88.07 | 96.05 | 92.26 | 48.57 | 97.5 | 85.64 | 91.02 |
| | mnasnet | 81.41 | 88.76 | 96.09 | 92.79 | 57.63 | 97.00 | 82.35 | 89.01 |
| | resnext50-32x4d | 83.42 | 87.32 | 95.37 | 95.97 | 52.87 | 96.65 | 83.37 | 91.74 |
| | wide-resnet50-2 | 84.30 | 87.40 | 95.16 | 95.85 | 57.77 | 96.74 | 76.23 | 90.22 |
| Cosine | resnet18 | 84.57 | 89.90 | 98.58 | 88.34 | 76.34 | 98.95 | 91.97 | 95.00 |
| L2 | resnet18 | 82.90 | 89.75 | 98.25 | 88.15 | 74.84 | 98.61 | 88.21 | 94.89 |
| NLL | resnet18 | **87.51** | **90.64** | **99.15** | 88.45 | **81.99** | **99.15** | **95.21** | **95.47** |

# E   Detailed Method

---

**Algorithm 1** Distillation through Gaussian Kernels

---

**Input:** Dataset $D = \{\mathbf{x}_i\}$, Embedders $(\mathsf{T}_k)_{1 \leqslant k \leqslant K}$, Student embedder $\mathsf{S}$, Number of iterations $T$, Learning rate $\eta$

Initialize the parameters $\theta_s$ of the student embedder $E_s$ and the parameters $\theta_k$ of the parametric Gaussian kernels

**for** $t = 1$ to $T$ **do**
   Sample a batch of inputs $\{\mathbf{x}_i\}$
   Compute the embeddings $\left\{\mathbf{t}_i^k = \mathsf{T}_k(\mathbf{x}_i)\right\}_{1 \leqslant k \leqslant K}$
   Compute the student embeddings $\{\mathbf{s}_i = \mathsf{S}(\mathbf{x}_i)\}$
   Compute the loss $\mathcal{L}_{NLL} = -\sum_{k=1}^{K} \sum_{i=1}^{N} \log \mathcal{N}(\mathbf{t}_i^k | \mu_k(\mathbf{s}_i), \Sigma_k(\mathbf{s}_i))$
   Update the parameters $\theta_s$ and $\theta_k$ using the Adam optimizer.
**end for**

---

# F   Computaional ressources

Our experiments were conducted in single GPUs settings. We used NVIDIA V100 GPUs for about 3000 GPUs hours to train our different models.

# G   Baselines

For the MSE, we will optimize the following loss function following SimReg strategy (Navaneet et al., 2022).

$$\mathcal{L}_{MSE} = -\sum_{k=1}^{K} \sum_{i=1}^{N} ||\mathsf{S}(\mathbf{x}_i) - \mathsf{T}_k(\mathbf{x}_i)||^2 \,, \tag{7}$$

where it calculates the summation of MSE between the representation produced by each teacher and the student, for each instance of the batch.

Variant of SimReg can be implemented for Cosine multi-teacher feature distillation(Gao et al., 2022; Navaneet et al., 2022), we optimize the summation of cosine of teachers and the students representations of each instance of the batch, *i.e.*:

$$\mathcal{L}_{Cosine} = -\sum_{k=1}^{K} \sum_{i=1}^{N} \frac{\mathsf{S}(\mathbf{x}_i).\mathsf{T}_k(\mathbf{x}_i)}{\max(||\mathsf{S}(\mathbf{x}_i)||_2 \,.\, ||\mathsf{T}_k(\mathbf{x}_i)||_2 \,, \epsilon)}. \tag{8}$$

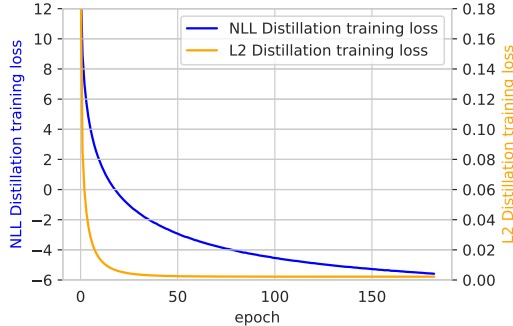

Figure 12: Training curves for the MSE baseline and the NLL student for the molecular experiments.

# H  Discussion On MSE distillation

We observed that when training with the MSE loss, the loss reaches a minimum in only a few epochs ( 40), but the distilled students achieve lower performances on downstream tasks. This could be due to the fact that the NLL loss is more expressive, and harder to optimize (see below). As a result the student learns more informative features compared to when trained with the MSE loss (Figure 12).

We can provide a theoretical insight to explain this phenomenon. Training using the negative log-likelihood over a Gaussian kernel is a simple generalization of the MSE. For a given multivariate Gaussian kernel parameterized by $\mu$ and $\Sigma$, we have:

$$- \log(p_{\mu, \Sigma}(x)) = \log(C) + \frac{1}{2} \log \det \Sigma + \frac{1}{2}(x - \mu)^T \Sigma^{-1}(x - \mu)$$

Minimizing the MSE loss boils down to minimizing this equation over only, with $\Sigma = I$. Therefore, minimizing the negative log-likelihood of a Gaussian kernel is strictly more expressive than minimizing the MSE directly, which could account for the performance gains we observe.


# I  Funding

This work was granted access to the HPC resources of IDRIS under the allocation AD011013290R3, and enabled by support provided by Calcul Quebec and the Digital Research Alliance of Canada. This work was funded through scholarships by "École de Technologie Supérieure Montreal", "Université Paris-Saclay" and "McGill University".

