# OpenReview forum: "Learning Task-Agnostic Representations through Multi-Teacher Distillation"
_NeurIPS.cc/2025/Conference — NeurIPS 2025 poster_

### Official Review · Reviewer_3Aq4 · 2025-06-04

**Clarity:** 2
**Significance:** 3
**Originality:** 2
**Rating:** 4
**Confidence:** 4

**Summary:**

This paper proposes a novel framework for task-agnostic multi-teacher distillation that leverages a majority vote loss formulation, theoretically grounded in mutual information between student and teacher embeddings. The authors develop a principled loss function using conditional entropy and demonstrate the approach across three domains: NLP, CV, and molecular modeling. Extensive experiments show that the distilled student models achieve competitive or superior performance compared to existing baselines of similar sizes, improving the Pareto frontier of model size vs performance.

**Questions:**

1. Have you considered adapting other feature-based distillation methods as task-agnostic baselines? For example, Correlation Congruence (ICCV 2019) and Relational Knowledge Distillation (CVPR 2019) could be easily applied in your setup by averaging over teacher features. Including them on a single domain like vision is enough to provide a more comprehensive comparison.

2. Could you analyze the stability of training compared to MSE or cosine baselines? Maybe you can provide convergence curves and variance across runs.

3. Could the proposed method be extended to cross-modal distillation, as hinted at in the conclusion? What challenges do you anticipate?

**Ethical Concerns:**

["NO or VERY MINOR ethics concerns only"]

**Final Justification:**

I appreciate the authors' response. The paper is overall well-executed, and the experimental section is quite thorough. That said, I believe that a score of 4 (borderline accept) remains the most suitable assessment.

**Limitations:**

Yes. The authors acknowledge that maximizing mutual information does not explicitly preserve the structure of the embedding space (e.g., cosine similarity), since information-theoretic objectives are invariant to invertible transformations. While this is theoretically valid, it does not appear to pose a practical problem in the reported experiments. The student models still achieve strong performance across tasks, suggesting that the learned representations remain effective despite the lack of explicit structure preservation. As such, I view this limitation as mostly theoretical and not impactful to the empirical validity or applicability of the method.

**Quality:**

3

**Strengths And Weaknesses:**

Strengths:

1. The paper connects Bayesian classifier disagreement bounds with conditional entropy and mutual information, offering a principled and task-agnostic loss formulation.

2. The method is validated on three diverse domains (text, vision, molecules), strengthening the claim of generality. Across multiple benchmarks (MTEB, TDC, fine-grained vision tasks), the distilled models consistently outperform or match larger models, pushing the size-performance tradeoff frontier. The comparison to standard MSE, cosine, and CompRess baselines is thorough and convincing.

Weaknesses:

1. While the proposed method is compared to MSE, cosine similarity, and CompRess, the set of baselines remains relatively narrow. There exist several established feature-distillation methods from the literature (e.g., Correlation Congruence for Knowledge Distillation [ICCV 2019]) that, while originally developed for task-specific distillation, can be straightforwardly adapted to a task-agnostic multi-teacher setup (e.g., by averaging teacher features). Including one or two such baselines (even only on a single modality like vision) would strengthen the empirical case and more convincingly isolate the advantages of the proposed information-theoretic loss.

2. Although the method is claimed to be more stable than MSE or cosine methods, the paper lacks deeper analysis (e.g., convergence curves, variance across runs) to support this point.

3. Additionally, there are some minor presentation issues. Several figures (e.g., in Section 4 and 5) are rasterized and appear blurry when zoomed in — using vector graphics would improve readability. There are also a few small typos and formatting inconsistencies: 1) Line 39: missing the “T” in “To our knowledge”; 2) Line 97: verb tense (“introduce”) is in past tense, inconsistent with surrounding text; 3) Line 176–177: the formatting of the letter “M” is inconsistent (italicized in one line but not the other). These issues do not affect the technical content but could be improved in the final version.

---

> ### Author Rebuttal · Authors · 2025-07-30
>
> We thank reviewer 3Aq4 for their thorough and insightful review. We appreciate that they found our method principled and our experimental setup thorough and convincing. We added additional baselines to our vision experiments and we are thankful to the reviewer for pointing out the unclear reference to the MSE training stability, as well as the mistaken reference in the conclusion. We will correct these in the revised version of the paper.
>
> ## MSE distillation stability
>
> We would like to clarify that our reference to the instability of Mean Squared Error (MSE) was drawn from previous works in the field of reinforcement learning \[1, 2, 3\], where stability is context-specific. **Our intention was to use this as motivational background for our work, rather than making a direct claim about stability within our own research. We acknowledge that this distinction was not clear in our initial presentation, and we will revise those references. Most importantly, we mistakenly referenced stability in our conclusion, which we will correct in the revised version.**
>
> *Additionally we will add a dedicated section comparing the MSE loss and our NLL loss (with the training curves):*
>
> *We observed that when training with the MSE loss, the loss reaches a minimum in only a few epochs (\~40), but the distilled students achieve lower performances on downstream tasks. This could be due to the fact that the NLL loss is more expressive, and harder to optimize (see below). As a result the student learns more informative features compared to when trained with the MSE loss. (training curves will be included)*
>
> *We can provide a theoretical insight to explain this phenomenon. Training using the negative log-likelihood over a Gaussian kernel is a simple generalization of the MSE. For a given multivariate Gaussian kernel parameterized by \\mu and \\Sigma, we have:*
>
> $\-log (p\_{\\mu, \\Sigma}(x)) \= \-\\log C \+ \\frac12 \\det \\Sigma \+ \\frac12 (x-\\mu)^T \\Sigma^{-1} (x-\\mu)$
>
> *Minimizing the MSE loss boils down to minimizing this equation over *$\\mu$* only, with *$ \\Sigma \= I$*. Therefore, minimizing the negative log-likelihood of a Gaussian kernel is strictly more expressive than minimizing the MSE directly, which could account for the performance gains we observe.*
>
> \[1\] Stop Regressing: Training Value Functions via Classification for Scalable Deep RL
> Jesse Farebrother and al. 2024
> \[2\] A Distributional Perspective on Reinforcement Learning, Marc G. Bellemare and al. 2017
> \[3\] Regression as Classification: Influence of Task Formulation on Neural Network Features
> Lawrence Stewart and al. 2022\.
>
> ## Extension to multi-modal setting
> **The method can indeed be extended to a multimodal setting, where modality specific encoders would share the same backbone to embed different types of data.**
> The challenge we would like to address is training a distillation model when limited cross-modal labels are available, to enable pretraining on larger datasets. So far, we did not observe a significant advantage compared to single modality training, and we are running additional experiments to answer the question: “How much cross-modal information is required to distill multi-modal representation that outperform the single modality ones?”
>
> ## Additional baselines
>
> We have added relational KD (RKD) \[Park et al., CVPR 2019\], Correlation Congruence with Gaussian RBF, and Bilinear (normalized features) kernels (CC-grbf, CC-Bilinear) \[Peng et al., ICCV 2019\] for vision. **As shown in the following table, our method (NLL) has better performance compared to the additional baselines.** Our intuition for the difference of accuracy is that both RKD and CC are proposed to work alongside the task loss, which could be an important signal for their optimization in practice.
>
> | Method | CIFAR10 | DTD | STL10 | SVHN | FGVCAircraft | CUB |
> | :---- | :---- | :---- | :---- | :---- | :---- | :---- |
> | RKD | 87.64 | 52.23 | 89.63 | 61.66 | 30.54 | 47.85 |
> | CC-grbf | 84.07 | 61.86 | 93.03 | 59.96 | 33.48 | 57.55 |
> | CC-Bilinear  | 92.95 | 61.22 | 95.42 | 63.71 | 35.16 | 64.7 |
> | NLL | **94.76** | **65.85** | **96.45** | **76.91** | **48.13** | **69.37** |
>
> ## Presentation Issues
> Thank you for your thoroughness. You are absolutely right, we’ll update the figures with vectorized versions, and we’ll fix the formatting inconsistencies you pointed out.
>
> Thank you again for your valuable feedback and suggestions, which have significantly improved our paper. Especially, thank you for pointing out the lack of clarity in our reference to the instability of MSE value estimation in reinforcement learning, that we have corrected.
> We hope we addressed all of your concerns so that you might consider raising the score of your review.

---

> > ### Comment · Reviewer_3Aq4 · 2025-08-01
> >
> > Thanks for the detailed rebuttal! You have responded thoroughly to most of my concerns.
> >
> > Just a quick question: do you expect the multimodal experiments to be ready within the next week? It’d be great to see even a preliminary result if feasible.

---

> > > ### Author Response · Authors · 2025-08-01
> > > **Preliminary Multi-Modal results**
> > >
> > > We are glad we answered most of reviewer 3Aq4’s feedback.
> > >
> > > We have begun extending our method to the multimodal setting with a focus on the medical domain, where unpaired  (i.e images and text are not assigned one to another), unlabeled or inconsistently labeled data is especially prevalent.
> > >
> > > For training, we used unpaired data from different modalities: histopathology image datasets for vision \[1, 2\], and medical textbook-style \[3\] corpora including for text. Our current setup comprises 3 teachers per modality We trained one multimodal student on both modalities to encourage knowledge coordination, inspired by work in coordinated representation learning, and one student trained only on the vision datasets.
> > >
> > > As an initial evaluation, we compared unimodal and multimodal training setups on downstream classification tasks using PCAM and CRC benchmarks. Results are as follows:
> > >
> > > | Modality | CRC | PCAM |
> > > | :---- | :---- | :---- |
> > > | Vision only | 95.91 | 85.95 |
> > > | Vision \+ text | 95.69 | 86.02 |
> > >
> > > **Overall the performances of the multimodal student are on par with the single modality student.** Hence, training a student to embed both image and vision does not seem to degrade the quality of the vision embeddings, but does not make them more informative for the moment.
> > >
> > > We are working on adding the cross-modal information, but unfortunately we don’t believe we will have results by this week for this.
> > >
> > > \[1\] Jewsbury, Robert, et al. "StainFuser: Controlling diffusion for faster neural style transfer in multi-gigapixel histology images." arXiv preprint arXiv:2403.09302 (2024).
> > > \[2\] Kather, J. N., Zöllner, F. G., Bianconi, F., Melchers, S. M., Schad, L. R., Gaiser, T., Marx, A., & Weis, C.-A. (2016). Collection of textures in colorectal cancer histology \[Data set\]. Zenodo.
> > > \[3\] Jin, D., Pan, E., Oufattole, N., Weng, W.-H., Fang, H., & Szolovits, P. (2021). What disease does this patient have? A large-scale open-domain question answering dataset from medical exams. Applied Sciences, 11(14).

---

> > > > ### Comment · Reviewer_3Aq4 · 2025-08-02
> > > >
> > > > I see. Thank you for your quick response!

---

### Official Review · Reviewer_zNKm · 2025-07-01

**Clarity:** 4
**Significance:** 3
**Originality:** 3
**Rating:** 5
**Confidence:** 3

**Summary:**

This paper presents a novel approach to Multi-Teacher Knowledge Distillation (KD), addressing the problem of generating general-purpose, task-agnostic embeddings.  Specifically, the core contribution is a task-enabling setting for multi-teacher distillation. Instead of traditional MSE-based losses, which can be unstable in high-dimensional spaces, the proposed method trains a student model to align its downstream task predictions with the collective predictions of an ensemble of teacher models. This is achieved through an ensembling loss that measures agreement between Bayesian predictors derived from student and teacher embeddings. A key theoretical finding is that this loss can be bounded independently of the specific task, utilizing the conditional differential entropy of the teachers' embeddings given the student's output, thereby providing a robust, task-agnostic student-teacher reconstruction loss. To evaluate the conditional entropy of the teachers’ embeddings given the student’s embedding, the authors propose using a parametric Gaussian model whose parameters are learned during the student’s training. Finally, the paper demonstrates high-quality generalized embedders across molecular modeling, natural language processing, and computer vision, with trained student models achieving competitive performance on a range of downstream tasks (e.g., classification, regression, clustering, sentence similarity).

**Questions:**

I feel the paper should be accepted (I gave it a 5) but I am not sure if there's anything that could be added for me to give a higher score.

**Ethical Concerns:**

["NO or VERY MINOR ethics concerns only"]

**Final Justification:**

Having read the authors' rebuttals and the other comments from the reviewers, I still maintain my score and positive opinion about this paper — which I feel that should be accepted.

**Limitations:**

Yes

**Paper Formatting Concerns:**

No conerns.

**Quality:**

3

**Strengths And Weaknesses:**

Strengths: This well-crafted paper proposes a fairly novel and interesting method, complemented by comprehensive and compelling experimental findings.

Weaknesses: No major weaknesses.

---

> ### Author Rebuttal · Authors · 2025-07-30
>
> We thank reviewer zNKm for their review, and we are glad they appreciated our work, and believe the paper should be accepted. We remain available to answer any new question if needed.

---

### Official Review · Reviewer_Aw1B · 2025-07-03

**Clarity:** 2
**Significance:** 3
**Originality:** 3
**Rating:** 4
**Confidence:** 3

**Summary:**

This paper proposes a multi-teacher distillation technique that is task agnostic. Through a majority vote objective function and ensembling loss, they show that this loss can be bounded independently of the task, making the distillation process task agnostic.

**Questions:**

please see weakness

**Ethical Concerns:**

["NO or VERY MINOR ethics concerns only"]

**Final Justification:**

I would like to keep the initial score. This paper addresses tiny models and from this discussion it's pretty clear that the technique proposed by the authors of this paper wont be beneficial on models that are >1B params. and even >700M params. So it casts doubts on the motivation for the paper and the relevance of the technique in light of the modern embedding models out there.

**Paper Formatting Concerns:**

no major concerns

**Quality:**

3

**Strengths And Weaknesses:**

Strengths:
- This paper formalizes the problem of task agnostic distillation well.
- The use of Gaussian mixture based estimator to formulate loss is quite interesting and novel
- They show a good understanding of the problem in designing embedding models, and show the relevance of this technique.
- The paper shows application of this method for distilling molecular embedders, which is an interesting and important application.

Weaknesses:
- The paper is a bit outdated.
- The paper does not motivate the need for task agnostic distillation well. A good comparison of modern embedding models on the benchmarks would have been useful, with more elaborate benchmarks.
- The existing evaluation has multi-teacher distillation techniques as the baseline, but overall, it should have newer embedding models in the baseline as well.
- the evaluations are done on really small models and small benchmarks.

---

> ### Author Rebuttal · Authors · 2025-07-30
>
> We thank reviewer Aw1B for their thorough review, and we are glad they appreciate the novelty in our proposed method. We politely disagree our experiments focused on 'really small models and small benchmarks', and we aim to provide a clear justification below.
>
> ## Small Models
> We trained models up to 300M parameters for textual embeddings, while this is not particularly large (<1B), **it is a standard size for embedding models in text**.
> Contrary to text generation where clear performance gain can be seen when using very large models, performances of embedding models in text can achieve very competitive scores, while having fewer parameters  (e.g., the Stella-500M model outperforming several 8B models on the MTEB).
> This scale is comparable to other well-established and widely used embedders, such as Stella and GIST, which are recognized for their strong performance in the field.
>
> ## Small benchmarks
>
> We politely disagree with this statement, across the three modalities evaluated in our study **we conducted comprehensive assessments on a total of 71 widely used datasets.** These included 6 datasets for computer vision, 32 for molecular modeling, and 33 for text analysis (ie **the Massive Text Embedding Benchmark, which constitutes the reference for textual embedders evaluation**).
>
> ## Motivation of the task-agnostic distillation
> In our introduction, we aim to motivate task-agnostic distillation in two steps. First, we highlight the development of embedding models, which are inherently task-agnostic. These models compress objects into numerical representations, thereby facilitating a wide array of downstream tasks. Next, we argue that the diversity of embedding models available in each field can be leveraged to build an embedder that benefits from these diverse representations through distillation.
>
> We will extend the paragraph motivating the use of embedding models by mentioning their current applications in a wide range of scenarios, including classification, clustering, and information retrieval. We will also emphasize their key advantage in terms of computational efficiency, particularly in scenarios with limited labeled data.
>
> ## 'The paper is a bit outdated'
>
> We are not sure what part of the paper you are referring to when you say it is a bit outdated. Could you be more specific so we can update and improve our work accordingly? Nevertheless, we would like to make a few remarks as the field of text embedding models have gained a lot of focus in research lately:
> - We chose our baselines based on their leading performance on the MTEB benchmark at the time of submission. Notably, **Stella 400m v5 was the top-performing model in its weight category, only recently surpassed by the Qwen embedders released on June 15th 2025.**
> - We compared our students against the best models in each category, **including GIST models, which were among the highest-performing embedders across most weight categories until early 2025.**
>
> In any case, we will update the MTEB benchmark tables in our paper to include the most recent results published.
>
> ## Choice of baselines
> **We did not only compare our approaches with multi-teacher distillation approaches.** While our analysis mainly focuses on the comparison of our method with multi-teacher distillation methods, we also compared its performances with several embedders for each modality (of the same weight category).
> **The objective of our experimental section is to validate that our distillation approach is efficient in compressing the information of several teachers into a smaller student,  thus our focus on distillation baselines.** It is hard to provide a fair comparison of methods when comparing with other embedders since they have been trained with widely different settings, datasets, infrastructures and training objectives, whereas we provide distillation comparison in a valid controlled setting for fair comparisons to answer our initial scientific question.
>
> We hope we have successfully addressed all your concerns, and we would be grateful if you could reconsider your score based on our revisions. Thank you again for your insightful review.

---

> ### Comment · Reviewer_Aw1B · 2025-07-31
>
> I thank the authors for their comments.
>
> My biggest point of disagreement so far is that this paper does not compare its approach with modern embedding models. And that is the reason why I mentioned that the paper might be a bit outdated, since the embedding models that the paper compares to are increasingly outdated. Even if the authors feel that comparing it with modern embedding models is not a fair comparison, it would still be good to have those experiments in the paper. And that might directly question the motivation of the paper. The question is, "In light of how well modern embedding models perform, do we really need a multi-teacher distillation approach to train embedding models? And if so, could you motivate it by including head-to-head comparisons with modern embedding models?"
>
> What's the harm in comparing it with models with a larger size and a larger context window?

---

> ### Author Response · Authors · 2025-08-01
> **Additional comparison with bigger models**
>
> We thank reviewer Aw1B for their involvement in the rebuttal process.
>
> **We want to insist that we do compare with the most recent/modern embedders of similar sizes at the time of the submission in the main part of the paper** (only the best one for each weight categories and we provide the full MTEB results in appendix C.2) and provide such a head-to-head comparison. And indeed, our method produces models that outperform all modern embedders in their size categories (for fair comparison), thus suggesting that the multiteacher setting indeed provides significant advantages.
> For comprehensiveness' sake, we include in this rebuttal the most recent results from the MTEB for the biggest/best models. (We had to trim the full table for it to fit in this answer, but we can provide any part of that huge table) compared to our own models. The most recent one is the Qwen 600M embedder, released only a few weeks ago.
>
> We agree there is no harm in comparing with larger models (we provide the results here as well) **however, it is important to keep in mind that it is not an apples-to-apples comparison. Models of different sizes and computational costs have different applications. Showing that we can achieve higher information density using our models has practical application for low-resource or on-edge deployment settings for which larger models are impractical.**
>
> Our medium model (109M) parameters are on par with models 5 times its size (Average performance 80.23), only outperformed by Stella 400M (still the best model of its category released early 2025 and included in the paper) and KALM (494M). **The best performing and most recent model by far is Qwen 600M, which only outperforms our models by 5 points.** Only models above 1B parameters achieve significant gains over our medium (109M parameters) model. **If you have any additional specific model in mind that you deem more recent, please let us know we will add it if it's present on the MTEB benchmark.
> We will update the final version of the paper with the most recent version of the MTEB.**
>
> Model                                       |#Params|EmbDim|AmazonCtf|Banking77|IMDB|MTOPDOMain|MassiveInt|MassiveScen.|ToxicConv.|TweetSent.|Average
> --------------------------------------------|-------|------|---------|---------|----|----------|----------|------------|----------|----------|-------
> Student-s-nll                               |32     |384   |77,3     |86,7     |88,3|95,5      |76,7      |80,7        |66,1      |60,6      |79,0
> Student-m-nll                               |109    |768   |79,6     |88,0     |88,3|96,2      |78,6      |82,7        |67,1      |61,3      |80,2
> stella_en_400M_v5                           |435    |4096  |94,3     |89,3     |96,5|98,3      |80,5      |89,6        |84,0      |73,6      |88,2
> KaLM-embedding-multilingual-mini-instruct-v1|494    |896   |81,5     |84,9     |95,0|92,2      |69,8      |74,2        |89,0      |76,5      |82,9
> KaLM-embedding-multilingual-mini-v1         |494    |896   |76,4     |79,2     |91,6|92,5      |70,9      |76,1        |70,8      |62,7      |77,5
> KaLM-embedding-multilingual-mini-instruct-v2|494    |896   |95,3     |89,5     |95,2|98,9      |77,8      |86,0        |89,3      |78,6      |88,8
> jina-embeddings-v3                          |572    |1024  |90,9     |84,1     |91,9|          |75,2      |84,1        |91,3      |71,4      |84,1
> snowflake-arctic-embed-l-v2,0               |568    |1024  |65,6     |81,8     |72,8|93,5      |71,5      |76,2        |65,9      |59,6      |73,4
> Qwen3-Embedding-0,6B                        |595    |1024  |91,5     |81,0     |95,4|96,0      |80,4      |83,6        |82,1      |76,0      |85,8
> stella_en_1.5B_v5                           |1500   |8960  |94,1     |89,8     |96,7|98,7      |84,5      |89,7        |86,8      |74,8      |89,4
> jasper_en_vision_language_v1                |1000   |8960  |93,8     |87,2     |97,0|99,2      |85,3      |91,2        |91,3      |77,2      |90,3
> Qwen3-Embedding-4B                          |4000   |2560  |93,7     |86,3     |97,2|97,8      |85,0      |88,8        |91,4      |78,4      |89,8

---

> ### Comment · Reviewer_Aw1B · 2025-08-05
>
> Thank you for the updated experiments. I would still like to keep my current score, as i believe that the role of multi-teacher distillation is increasingly being replaced by more powerful models. While the authors focus specifically on tiny to small models, that are sub 0.5 to 1 B parameters (mostly sub 0.5B params), the SOTA embedding models are largely > 1B params. The cost of using these models is already very low and further going down. Further, it can be safely extrapolated from the current table that almost all, if not all, models of size > 1B params will outperform the Multi-Teacher Distillation strategy presented in this paper.

---

> > ### Author Response · Authors · 2025-08-09
> >
> > We would like to thank reviewer Aw1b for their reviews and engagement in the rebuttal process.
> >
> > We believe that to assess the value of multi-teacher distillation to train embedding models, models of similar sizes should be compared for a fair comparison (ideally in a controlled setting).
> >
> > To compare with these larger models, a larger student should be trained. We acknowledge this is a limitation of our work (and of distillation that often aims to distill large models into smaller ones) and we will discuss it in the limitation section of the revised version of the paper.
> >
> > We thank again reviewer Aw1b for their reviews, engagement along this rebuttal period, and despite our disagreements we completely respect their final decision.

---

> > > ### Comment · Reviewer_Aw1B · 2025-08-09
> > >
> > > Yes, distillation is from larger to smaller. But this paper deals with tiny models. I am just curious about how models that are of the order of a billion parameters fare with this methodology. I believe it's unfair to say that it's a limitation of distillation to be able to compare larger models than the ones that this paper studies.

---

> > > > ### Author Response · Authors · 2025-08-09
> > > >
> > > > We agree it would have been interesting to experiment with >=1B embedders.
> > > >
> > > > To perform these experiments, we would have had to use large embedders which were, at the time of the submission, not as performant relative to their size (Qwen3 and KALM models being released a month after this submission).
> > > >
> > > > Besides, this would have required further resources we unfortunately didn't have to expand the text experiments.
> > > >
> > > > We will discuss this limitation (comparison to these new embedders) in the revised version of the paper.
> > > >
> > > > We thank reviewer Aw1b once more for his continuous feedback.

---

### Official Review · Reviewer_dFG5 · 2025-07-03

**Clarity:** 4
**Significance:** 4
**Originality:** 3
**Rating:** 5
**Confidence:** 4

**Summary:**

This paper proposes a task-agnostic framework for multi-teacher distillation that learns general-purpose representations without requiring task-specific labels or supervision.
The approach introduces a novel loss function based on a “majority vote” principle, which is shown to be bounded by the mutual information between the student and teacher embeddings.
This results in a task-agnostic objective that encourages the student to align with the ensemble of teachers across a wide range of potential downstream tasks.

The method leverages a differentiable, Gaussian mixture-based estimator of conditional entropy to implement this loss in practice.
The training procedure minimizes the negative log-likelihood of the teacher embeddings conditioned on the student’s output, allowing end-to-end learning of a compact, informative student embedder.

The paper evaluates the approach across three domains - natural language processing, computer vision, and molecular modeling - using a range of classification, regression, and clustering tasks.
The results demonstrate that the distilled student models achieve competitive or superior performance relative to both teacher models and size-matched baselines.
The distilled model by the proposed method also shows strong size-performance trade-offs, advancing the Pareto frontier for efficient representation learning.

**Questions:**

1. Despite the strengths of the work, there is no dedicated discussion of its limitations, either theoretical or practical. For example, the method may face challenges in embedding structure preservation (as the authors briefly mentioned), memory costs from storing teacher embeddings, or reliance on high-quality teachers. Could the authors add an explicit limitations section/paragraph discussing practical boundaries of the approach, including cases where it may underperform or become inefficient? Can the authors elaborate on the practical implications of unstructured embeddings? Have the authors observed cases where this limitation actually causes issues?
2. It may also be related to the above - Line 253, Page 8: Have any reasoning behind why the default 8-teachers model struggles and performs worse than other 1 or 2-teachers models with the BBB (Distribution) task?

Minor issues
- Line 39, Page 2: “o our knowledge, …”
- No caption for Figure (probably) 4

**Ethical Concerns:**

["NO or VERY MINOR ethics concerns only"]

**Final Justification:**

This paper proposed a novel approach for task-agnostic multi-teacher distillation method.
The paper was well written, while the authors addressed the concerns raised, mainly about discussion of limitations.
The paper should be accepted so that the community benefits from the work and extend it.

**Limitations:**

Please refer to the weaknesses and the questions above.

**Quality:**

3

**Strengths And Weaknesses:**

**Strengths**
- The paper introduces a theoretically grounded, task-agnostic distillation objective based on minimizing the conditional entropy of teacher embeddings given the student’s. This formulation allows the student to learn diverse, informative representations without reliance on task-specific labels, offering a general and conceptually sound approach.
- The method is validated across three domains - language, vision, and molecular modeling - demonstrating its versatility. The breadth of evaluation strengthens the claim that the learned representations are useful across a wide range of downstream tasks.
- The distilled student models consistently achieve high performance for their parameter count. In many cases, they match or outperform significantly larger models, indicating the method’s effectiveness in capturing information from multiple teachers.

**Weaknesses**
- While the authors acknowledge that the proposed objective does not preserve structural relationships (e.g., cosine similarity) in the embedding space, which helps clarify the scope and nature of the learned representations, the paper does not include a dedicated discussion of the limitations of the method in general. For instance, the handling of teacher inconsistency, conditions where the method may underperform, etc., are not discussed.

---

> ### Author Rebuttal · Authors · 2025-07-30
>
> We thank reviewer dFG5 for their insightful review and are pleased that they found our work interesting. Following the reviewer's advice, we will include a section dedicated to the limitations of our approach, referencing some results of other sections.
>
> ## Limitation discussions
>
> **While we discussed some limitations in the appendices and experimental section, we agree our work would benefit from a dedicated limitation section. We will add a specific section covering the following limitations of our method:**
> - **Application Scope:** Our method develops student embedding models primarily for diverse, unknown tasks. For single pre-defined tasks, task-specific distillation approaches might be more suitable.
> - **Overhead in Distillation:** Like any distillation setting, especially multi-teacher distillation, there is an overhead due to the distillation of large teachers into a smaller model. This overhead can be computational (if teacher embeddings are obtained by running inference for each training) or memory-related (if teacher outputs are precomputed and used during training). We opted for the latter as it drastically speeds up student training, initially storing all necessary embeddings on disk. For text (our most computationally demanding application), this amounts to approximately 100GB of embeddings for the largest teacher.
> - **Teacher Quality:** Our approach requires high-quality teachers relevant to the downstream tasks. In Section D.4, Table 26, we explore the impact of training a student embedder with task-specific teachers (classification, object detection and segmentation). We demonstrate that while task-specific teachers may offer limited benefits outside their domain, they do not negatively impact the students’ learning when used alongside task-relevant teachers.
> - **Structural relationship:** As pointed out in Section 4.2, our metric only optimizes the mutual information between the student and the teachers, it does not directly enforce any structure on the embedding space, which could harm the performances of our models for clustering tasks for instance. For textual embeddings benchmark, we observe clear gains in classification tasks (where a small classifier is trained on top of the embeddings), but the gains are less clear for clustering and STS tasks that rely on the dot product between embeddings to assess text similarity (See full results in App. C.2.2).
> - **Representative dataset:** To effectively embed data for future tasks, our method requires a training set that is representative of the data distributions of these tasks. This limitation, however, is common to all embedding models, which all require a diverse and representative dataset for training.
>
>
> ## Comparison of 8-teachers to 2-teachers on the BBB benchmark.
>
> The student trained in molecular modeling with 8 teachers indeed shows slightly lower performance on the BBB (blood-brain barrier) benchmark compared to students trained with 1 or 2 teachers.
> The BBB benchmark data distribution significantly differs from our training set, representing a domain shift compared to the training set of the students.
> Furthermore, it is one of the benchmarks where teacher performance is most tightly packed, with variations within 1.45 times the average standard deviation of the results. This could explain why training with 8 teachers performs closely to 1 or 2 (the differences in AUROC being half the standard variation), as all teachers demonstrate comparable performance on this specific task.
> We believe this explains the slightly lower average performance of the 8-teacher student compared to the 1 or 2-teacher students.
>
> Thank you again for your valuable feedback. We believe these clarifications and updates will address your concerns and enhance the quality and relevance of our work. We remain available to clarify any additional points if needed.

---

> > ### Comment · Reviewer_dFG5 · 2025-08-04
> >
> > Thank you for addressing the concerns that I had and asked. I hope the discussion about the limitations as well as the BBB result are included in the revised paper. Also, please correct and update minor issues, including typos and missing figure captions.

---

### Note · Authors · 2025-08-12

We thank all reviewers for their
 constructive feedback, and engagement during the discussion phase.

We appreciate that the reviewers recognized the originality and sound theoretical basis of our task-agnostic multi-teacher distillation approach, as well as their positive assessment of its strong empirical results across multiple modalities.

As requested by the reviewers:

* In the revised manuscript, we have added a dedicated section outlining limitations of our method.
* We clarified the BBB result in molecular modeling, attributing its behavior to domain shift and tight teacher performance variance.
* We added additional distillation baselines such as Relational KD and Correlation Congruence (with Gaussian RBF and Bilinear kernels), showing our method’s consistent advantage
* We clarified and revised the context of MSE “instability” references.
* We initiated multimodal experiments in the medical domain, which we aim to explore in future work in future work.
* We will update the results with the most recent results on the MTEB benchmark. Although larger models can achieve higher absolute performance, our students still deliver competitive accuracy at a fraction of the size and cost of these larger embedders. We added a discussion on the comparison with these larger and newer embedders that, although released in the last two month, we believe is relevant to this work.
* We have corrected typos, improved figure quality with vector graphics, and ensured consistency in formatting.

We thank once more and for last all reviewers for their engagement during this period, and for their feedback on our submission.

---

### Decision · Program_Chairs · 2025-09-17

**Decision:**

Accept (poster)

**Comment:**

This paper proposes a task-agnostic method for multi-teacher distillation utilizing a newly proposed loss function which is bounded resulting in a task agnostic objective. All reviewers agree that this is a paper with good methodological contribution and novelty (which is rare for the fairly saturated topic of knowledge distillation), and strong results across multiple datasets and domains. The authors did a good job with rebuttal addressing most of the authors' concerns. There is only some issue raised pertinent to what extent the method can be used for distilling very big models but the authors have already shown results with a 300M model, so this limitation can be considered only minor. Overall, a clear accept.